# Comparing the impact of environmental conditions and microphysics on the forecast uncertainty of deep convective clouds and hail

Constanze Wellmann[1], Andrew I. Barrett[1], Jill S. Johnson[2], Michael Kunz[1], Bernhard Vogel[1], Ken S. Carslaw[2], and Corinna Hoose[1]

[1]Institute of Meteorology and Climate Research, Karlsruhe Institute of Technology, Karlsruhe, Germany
[2]Institute for Climate and Atmospheric Science, School of Earth and Environment, University of Leeds, Leeds, UK

**Correspondence:** Constanze Wellmann (constanze.wellmann@alumni.kit.edu) and Corinna Hoose (corinna.hoose@kit.edu)

**Abstract.** Severe hailstorms have the potential to damage buildings and crops. However, important processes for the prediction of hailstorms are insufficiently represented in operational weather forecast models. Therefore, our goal is to identify model input parameters describing environmental conditions and cloud microphysics, such as vertical wind shear and strength of ice multiplication, which lead to large uncertainties in the prediction of deep convective clouds and precipitation. We conduct a comprehensive sensitivity analysis simulating deep convective clouds in an idealized setup of a cloud-resolving model. We use statistical emulation and variance-based sensitivity analysis to enable a Monte Carlo sampling of the model outputs across the multi-dimensional parameter space. The results show that the model dynamical and microphysical properties are sensitive to both the environmental and microphysical uncertainties in the model. The microphysical parameters, lead to larger uncertainties in the output of integrated hydrometeor mass contents and precipitation variables. In particular, the uncertainty in the fall velocities of graupel and hail account for more than 65% of the variance of all considered precipitation variables and for 30-90% of the variance of the integrated hydrometeor mass contents. In contrast, variations in the environmental parameters — the range of which is limited to represent model uncertainty — mainly affect the vertical profiles of the diabatic heating rates.

## 1 Introduction

Due to the large damage potential associated with severe convective storms, the forecast of deep convective clouds should be as accurate as possible. Thus, numerous studies have been published on simulating deep convective clouds. These have investigated how environmental parameters like wind shear (e.g., Weisman and Klemp, 1984; Lee et al., 2008; Fan et al., 2009; Chen et al., 2015; Dennis and Kumjian, 2017), and the aerosol environment, which determines the CCN concentration (e.g., Lee et al., 2008; Rosenfeld et al., 2008; Fan et al., 2013), affect the clouds in these simulations.

In Wellmann et al. (2018) we investigated the impact of simultaneous variations of six parameters describing environmental conditions. These parameters include CCN and INP concentrations, wind shear, thermodynamic profiles and two parameters characterizing the trigger mechanism used to initiate convection. The results showed that integrated hydrometeor mass contents and precipitation are most sensitive to variations of the CCN concentration and the vertical temperature profile. Moreover, different mechanisms for artificially triggering convection (a warm bubble, a cold pool or a bell-shaped mountain ridge) are compared revealing that the sensitivities depend on the choice of the trigger.

In addition to thermodynamic profiles and environmental conditions determining the formation and structure of deep convective clouds, also microphysical parameterizations have been shown to play a role. White et al. (2017), for example, simulated three cloud types using the Morrison (Morrison et al., 2005, 2009; Morrison and Milbrandt, 2011) and the Thompson (Thompson et al., 2004, 2008) bulk microphysics schemes varying the cloud droplet number concentration. They found that the use of the two schemes causes larger differences than the changes in the number concentration, primarily because of the representation of autoconversion of cloud water to rain and of cloud ice to snow. Splinters of ice particles, which can be generated during the riming process, favor the growth of ice from both the vapor and liquid phase because of their crystal lattice structure (Houze, 1993). This process of secondary ice production was introduced by Hallett and Mossop (1974) and is thus referred to as the Hallett-Mossop process. Connolly et al. (2006) simulated a thunderstorm over northern Australia to examine the impact of CCN and INP concentrations including variations of the strength of the Hallett-Mossop process. The results show that the height of the cloud top depends on the strength of the Hallett-Mossop process, whereas the mean precipitation is rather insensitive to these changes. In Johnson et al. (2015) the sensitivity of twelve deep convective cloud properties to uncertainties in nine microphysical processes was studied in a spectral bin microphysics model, using an emulator approach. They found that the cloud properties, including accumulated precipitation and maximum precipitation rates, are sensitive to a combination of aerosol concentrations and microphysical assumptions in the model.

Additional relevant parameters are the size distributions and the fall speeds of hydrometeors. Igel and van den Heever (2017b) varied the shape parameter of the cloud droplet size distribution in simulations of non-precipitating shallow cumulus clouds. They noticed an impact of this variation on the cloud droplet number concentration, the droplet diameter and the cloud fraction. They found that some of these effects are on the same order of magnitude as aerosol effects. Adams-Selin et al. (2013) investigated the effect of graupel size and thus also of the fall speed on deep convection. Their results show that "hail-like" (large and dense, with a high fall velocity) graupel immediately falls out of the cloud, leading to a reduced convection intensity. In contrast, smaller and slower falling graupel particles stay longer in the cloud, which results in more persistent convection. Also the results of Johnson et al. (2015) indicate that the fall speed of graupel is an important parameter influencing the precipitation rate. Moreover, field study observations indicate that hydrometeors may have a broad range of fall velocities (Knight and Heymsfield, 1983; Yuter et al., 2006; Heymsfield et al., 2018), which implies that there is large uncertainty in the result of the model parameterizations of the fall speeds. Gilmore et al. (2004) and Posselt and Vukicevic (2010) varied both the fall speeds and the densities of hail/graupel and snow, and found that these parameters impact the amount of precipitation significantly.

The development of deep convective clouds is sensitive to both environmental conditions and model parameters, but these

sensitivities are usually examined separately. A few studies, including Lee et al. (2008) and Storer et al. (2010), have analyzed the effect of several parameters, yet the maximum number of considered parameters is three or less. In this study, we combine various parameters related to both environmental conditions and microphysics into a single comprehensive sensitivity analysis. In idealized high-resolution model simulations, the selected input parameters are modified and their effect on the model output is analyzed with a special focus on precipitation and thermodynamic quantities. To our knowledge, the only previous studies of multiple (six or more) interacting uncertainties in deep convective clouds are our own previous studies (Johnson et al., 2015; Wellmann et al., 2018).

In general, the approach usually applied for the analysis of the sensitivity of the model output to changing input parameters is to vary a chosen parameter in a given range while other parameters are kept constant. This so-called *one-at-a-time* (OAT) analysis is applicable if the impact of a single model input is of interest. However, not only the effect of each input parameter independently will be assessed in this study, but also the relative contribution of the input parameters and their interactions to the whole uncertainty of the output is of interest. In reality, severe convective storms form in a wide range of ambient conditions, where either thermodynamic conditions or dynamic conditions may be the main driver, leading to different organizational forms of the storms. The gradual and combined variation of various parameters better represents real conditions compared to the OAT approach. To achieve this, we apply the methods of statistical emulation (O'Hagan, 2004, 2006) and variance-based sensitivity analysis (Saltelli, 2008), where the uncertainty of the output is densely sampled and then decomposed into contributions from the individual model input parameters while simultaneously considering their interactions. Thereby the relative contributions of each parameter to the uncertainty of the output can be quantified. The applicability of this approach for complex atmospheric models is demonstrated in Lee et al. (2013) and Johnson et al. (2015). Wellmann et al. (2018) also used this approach to investigate how environmental conditions impact the model output when simulating deep convective clouds. They quantify the contributions of parameters describing environmental conditions to the uncertainties of the integrated hydrometeor mass contents, precipitation and the size distribution of surface hail. In addition, the emulators are used to examine the sensitivity to changing CCN concentrations in different regimes of environmental conditions and the results are compared for three trigger mechanisms of deep convection, i.e. a warm bubble, cold pool and orography.

Here, we focus on the warm bubble as the trigger mechanism as it is frequently used in idealized studies, but we extend the set of uncertain input parameters to include not only environmental conditions but also microphysical parameters. Consequently, we compare the impact of environmental conditions and microphysics to quantify the individual contributions of the various parameters to the forecast uncertainty of precipitation-related quantities including hail. We also consider the vertical profiles of the diabatic heating rates in our analysis. This analysis and the choice of output variables are based on the results of the first author's PhD thesis (Wellmann, 2019) wherein more detailed descriptions are given.

A general description of the model setup and the input parameters is given in section 2, followed by an explanation of the methods of statistical emulation and variance-based sensitivity analysis in section 3. The considered output variables are described in section 4 and the results of the sensitivity analyses are presented in section 5. Conclusions are found in section 6.

## 2 Model Setup

For the simulations in this study, the limited-area numerical weather prediction model COSMO (Consortium for Small-Scale Modeling) (Baldauf et al., 2011; Schättler et al., 2016) developed by Deutscher Wetterdienst (DWD) and the COSMO consortium is used. Identical to Wellmann et al. (2018), we run COSMO in a convection-resolving idealized setup covering a domain of $700 \times 500$ grid points with a horizontal grid spacing of 1 km. This grid spacing was shown to be sufficient for the simulation of precipitation and hydrometeor mass content of idealized supercells, although vertical transport and timing differ from simulation at higher resolutions (Potvin and Flora, 2015; Huang et al., 2018). There are 64 vertical levels extending to a height of 23 km. These levels follow the transformation given in Gal-Chen and Somerville (1975) such that they are denser near the ground and further apart with increasing height (approximately 300 m vertical distance at 5 km altitude and 400 m vertical distance at 10 km altitude). Variables are written out and analysed on interpolated z-levels with 250 m vertical distance up to 3 km and 500 m vertical distance above. Open boundary conditions are used to prevent a simulated hailstorm from influencing itself via reflection at the boundaries. Moreover, we switch off the radiation scheme and neglect the Coriolis force in the simulations. The initial temperature and humidity profiles (which are also used when air is advected into the domain through the boundaries) are based on those of Weisman and Klemp (1982) to maintain atmospheric conditions favoring the development of deep convection. According to their profile, the maximum specific humidity $q_{v0}$ is chosen to be $12\,\mathrm{g\,kg^{-1}}$ at the lowest level. The vertical wind profile is comparable to the hodograph of quarter-circle shear introduced by Weisman and Rotunno (2000). Furthermore, the model uses the two-moment bulk microphysics scheme by Seifert and Beheng (2006), including a saturation adjustment approach (i.e. bringing relative humidity back to exactly 100% within one time step when supersaturation with respect to water occurs), predicting both the mass mixing ratios and the number densities of six hydrometeor classes (cloud droplets, rain, cloud ice, snow, graupel and hail). In our simulations, deep convection is triggered by a warm bubble as this mechanism is widely used in atmospheric modeling. The bubble is released at $\Delta x = 80$ km and $\Delta y = 200$ km at model initialization. We run the simulations for six hours with a time step of $\Delta t = 6$ s, where the first hour of the simulations is regarded as spin up and thus excluded from the analysis. During this simulation period, the clouds do not reach the boundaries of the domain. We consider only cloudy grid points (where the vertically integrated mass content of any hydrometeor type is $> 0$) in our analysis of the vertically integrated hydrometeor mass contents. Exemplary vertical and horizontal cross sections of the idealized convective cloud simulated with this configuration are shown in Wellmann et al. (2018, their Fig. 3). Typically, the cloud contains more graupel than hail at upper levels, but hail persists longer below the melting level and (in addition to rain) only hail, not graupel, reaches the ground.

We have taken a staged approach to our analysis of the effects of uncertain inputs on model output uncertainty for COSMO. We first explored the effects of the environmental conditions (section 2.1), and the full analysis for this study is given in Wellmann et al. (2018). Building on this work, we used the same approach to consider the corresponding effects of microphysical parameters in isolation (section 2.2). We then constructed a further final ensemble (section 2.3) using only the key inputs of

the setup with variation of environmental conditions and the new setup with variations in microphysical parameters, in order to enable a comparison of the relative importance of environmental and microphysical uncertainties for model output uncertainty. Note that as the results depend crucially on the ranges over which the parameters are varied, these have to be chosen carefully and taken into account when comparing to other studies.

## 2.1 Setup 1 - Varying environmental conditions

The input parameters of interest in this study are assigned to either describe environmental conditions, microphysics or both, where the parameter ranges relate to observations and model uncertainty. Regarding the environmental conditions, CCN concentration, INP concentration, wind shear, vertical temperature profile, and characteristics of the warm bubble, in terms of temperature perturbation and horizontal radius, are perturbed. An overview of these parameters and their respective ranges is given in Table 1. These parameters are referred to as *Setup 1* (S1).

**Table 1.** Overview of the uncertain input parameters and their ranges regarding environmental conditions (Setup 1). The parameters marked by * are included in Setup 3 which combines environmental conditions and microphysical parameters.

| input | min | max | units |
|---|---|---|---|
| CCN concentration * | 100 | 4000 | $cm^{-3}$ |
| INP concentration * | 0.01 | 10 | scaling factor |
| wind shear ($F_{shear}$) * | 0.3333 | 0.6666 | scaling factor |
| potential temperature at the ground $\theta_0$ (WK profile) * | 299 | 301 | K |
| temperature perturbation $\Delta T$ | 2 | 5 | K |
| horizontal radius $R_{hor}$ | 5 | 15 | km |

CCN, essential for the formation of cloud droplets, affect the dynamics and microphysics of the clouds (Rosenfeld et al., 2008; Tao et al., 2012; Fan et al., 2013, e.g.,). The cloud droplet activation scheme implemented in COSMO is based on grid-scale supersaturation and empirical power law activation spectra and uses look-up tables introduced by Segal and Khain (2006). Moreover, the vertical profile of the aerosol concentration has its maximum in the lowest 2 km above the ground and follows an exponential decrease with a scale height of 1 km towards higher altitudes. We vary the maximum CCN concentration between $100\,cm^{-3}$ and $4000\,cm^{-3}$ simulating both clean and polluted conditions. INPs affect the number of ice particles in the cloud as they support the formation of cloud ice (Houze, 1993), comparable to CCN generating cloud droplets. For INP changes, a scaling factor is applied to three microphysical processes. These processes are the deposition nucleation of cloud ice, the immersion freezing of cloud droplets and the immersion freezing of rain. The heterogeneous ice nucleation scheme of Huffman and Vali (1973) is implemented for the formation of cloud ice, while a stochastic model following the measurements of Bigg (1953) is used for the freezing of cloud droplets and rain. In this study, the scaling factor is varied between 0.01 and

10 on a logarithmic scale. This range is chosen according to DeMott et al. (2010) representing the range of INP concentrations measured in different field campaigns. We apply the same value of the scaling factor to all three processes.

According to several observational and modeling studies, directional shear is most important for the organization of convection (Weisman and Rotunno, 2000; Davies-Jones, 2015; Dennis and Kumjian, 2017). Therefore, we choose the initial vertical profile of the wind velocity to be constant in all simulations, whereas a scaling factor $F_{shear}$ determines the initial vertical profile of the wind direction ($WD$):

$$WD(z) = \begin{cases} 270° - F_{shear} \cdot 90°(1 + \frac{z}{6000 \text{ m}}) & ,z \leq 6000 \text{ m} \\ 270° & ,z > 6000 \text{ m} \end{cases} \tag{1}$$

Depending on the choice of $F_{shear}$, the wind direction near the ground is set. It linearly turns towards western directions with increasing height until a straight westerly flow is reached at a height of 6 km. For example, $F_{shear} = 0$ represents westerly wind at all heights and $F_{shear} = 1$ specifies southerly wind near the ground. Here, we vary $F_{shear}$ only between 0.3333 and 0.6666, corresponding to a wind direction at the ground between 210° and 240°, which reflects the typical error range of the operational COSMO forecast of the wind direction (Felix Fundel, personal communication, 2017).

The vertical profile of the potential temperature is implemented according to Weisman and Klemp (1982):

$$\theta(z) = \begin{cases} \theta_0 + (\theta_{tr}\theta_0) \left(\frac{z}{z_{tr}}\right)^{5/4} & ,z \leq z_{tr} \\ \theta_0 \exp\left(\frac{g}{c_p T_{tr}}(z - z_{tr})\right) & ,z > z_{tr} \end{cases} \tag{2}$$

It is based on the near-surface potential temperature $\theta_0$ initially set to 300 K, along with the tropopause height $z_{tr}$ and the tropopause temperature $T_{tr}$. In our study, $\theta_0$ takes values between 299 K and 301 K representing the typical error range of the operational temperature forecast of the COSMO model (Felix Fundel, personal communication, 2017). This variation of $\theta_0$ impacts the entire tropospheric profile and corresponds to a change of the convective available potential energy (CAPE) from 1210 J kg$^{-1}$ to 1347 J kg$^{-1}$.

The warm bubble is characterized by a temperature perturbation $\Delta T$ and a radius $R_{hor}$. Its maximum temperature perturbation $\Delta T$ is located in the center of the bubble and varies between 2 K and 5 K. The horizontal radius ranges between $R_{hor} = 5$ km and $R_{hor} = 15$ km, while the vertical extent is fixed at $R_z = 1400$ m. The variation of $\Delta T$ and the radius alter the strength of the trigger as different buoyancy gradients arise.

As the wind shear and the temperature are part of the operational forecast, their parameter ranges are the only ones that can be related to typical forecast errors. The ranges of the remaining parameters cover a wide variety of atmospheric conditions since there is no information from a forecast. These specifications are identical to those of the sensitivity analysis related to typical forecast errors in Wellmann et al. (2018).

## 2.2 Setup 2 - Varying microphysical parameters

The microphysical parameters analyzed in Setup 2 (S2) are the fall velocities of rain, graupel and hail, the strength of the ice multiplication and the shape parameter of the size distribution of cloud droplets. In addition, the CCN and INP concentrations are included in this set of input parameters. Table 2 summarizes the input parameters of Setup 2 and their considered ranges.

**Table 2.** Overview of the uncertain input parameters and their ranges regarding cloud microphysics (Setup 2). The parameters marked by [*] are included in Setup 3 which combines environmental conditions and microphysical parameters.

| input | min | max | units |
|---|---|---|---|
| CCN concentration [*] | 100 | 4000 | $cm^{-3}$ |
| INP concentration [*] | 0.01 | 10 | scaling factor |
| fall velocity of rain $(a_R)$ | 0.3 | 1.7 | scaling factor |
| fall velocity of graupel $(a_G)$[*] | 0.3 | 1.7 | scaling factor |
| fall velocity of hail $(a_H)$[*] | 0.7 | 1.3 | scaling factor |
| ice multiplication | $0.1 \cdot 10^8$ | $7 \cdot 10^8$ | $kg^{-1}$ |
| shape parameter | 0 | 8 | - |

The fall velocities of the precipitating hydrometers rain, graupel and hail are implemented in the model following mainly empirical equations based on measurements that describe the relation between the size or other characteristics of the particles and their fall velocities (Locatelli and Hobbs, 1974; Knight and Heymsfield, 1983). This uncertainty propagates in the microphysics scheme as the fall velocity impacts collision processes such as accretion and riming. To assess the uncertainty,
scaling factors are multiplied with the fall velocities of rain $(a_R)$, graupel $(a_G)$ and hail $(a_H)$. The ranges of the scaling factors are chosen based on the measurements of Yuter et al. (2006) and Knight and Heymsfield (1983) which suggest a spread of about 70% around the mean of the fall velocities of rain and graupel and a spread of about 30% of the fall velocity of hail, respectively. The production of ice splinters during the riming process introduced by Hallett and Mossop (1974) is a source of secondary ice particles. As their measurements show a large spread (Hallett and Mossop (1974), Fig. 2), we vary the splintering
coefficient in the COSMO model describing the number of secondary ice particles per kg rime between $0.1 \cdot 10^8$ $kg^{-1}$ and $7 \cdot 10^8$ $kg^{-1}$ to represent the range of their measurements. The size distribution of the hydrometeors has a substantial impact as various microphysical processes such as condensation or sedimentation depend on this. Thus, uncertainties in the size distributions have several possibilities to affect the processes in the microphysics scheme. By modifying the shape parameter of the cloud droplet size distribution, we assess the variation of the model output due to these input uncertainties. In the two-moment
scheme of COSMO, the size of the cloud droplets is described by a generalized $\Gamma$-distribution (Seifert and Beheng, 2006), where $\mu$ and $\nu$ are shape parameters of the distribution (see also Section 4.3). The default values are $\mu = 0.3333$ and $\nu = 0.0$, respectively. Here, $\mu$ is kept at its initial value, while $\nu$ is varied between $0$ and $8$ similar to Igel and van den Heever (2017a, b) who based their choice on the results of several measurement campaigns. This variation of the shape parameter changes the size distribution between broad distributions with lower number concentrations and narrow distributions with higher number
concentrations.

## 2.3 Setup 3 - Combined varying environmental conditions and microphysical parameters

Based on the results of the sensitivity analysis for hydrometeor and precipitation variables in setups S1 and S2, where the sets of environmental conditions and the cloud microphysics parameters are treated separately (Fig. 5 of Wellmann et al. (2018) and Fig. 1 of this manuscript), the input parameters of this combined Setup 3 (S3) are chosen such that the most important parameters of both environmental conditions and microphysics (those that contribute most to output uncertainty across the selected output variables) are considered in addition to the CCN and INP concentrations. The less important input parameters of S1 and S2 have not been reconsidered in order to limit the computational effort for conducting S3. For the variations of the environmental conditions, the parameters identified to affect the uncertainty most are the vertical wind shear and the potential temperature $\theta_0$ (Wellmann et al., 2018). The relevant parameters of the microphysics setup are the fall velocity of graupel and the fall velocity of hail (section 4). Detailed descriptions of these input parameters were already given in sections 2.1 and 2.2, and the same parameter ranges are used. The parameters included in S3 are marked by $*$ in Tables 1 and 2.

## 3 Methods

We identify the parameters leading to the uncertainty in each model output via a variance-based approach, which is a global sensitivity analysis meaning that all of the multi-dimensional parameter space is sampled (Saltelli, 2008).The output uncertainty is decomposed into contributions from each input parameter individually and also contributions from interactions of the parameters (see section 3.2). However, a large number of simulations is required to infer those contributions, which is not feasible for a complex numerical weather prediction model such as COSMO because of the high computational cost. Instead, we employ the approach of statistical emulation to build a surrogate model based on a set of training data. The emulator represents the relationship between a set of input parameters and a specific model output substantially reducing the number of model runs required to generate the data necessary for the variance-based sensitivity analysis. The following two sections give a summary of the emulator approach using Gaussian processes and the variance-based sensitivity analysis. More detailed descriptions of these methods are given in O'Hagan (2004, 2006); Saltelli et al. (1999); Johnson et al. (2015) and Wellmann et al. (2018).

### 3.1 Gaussian process emulation

First, a set of uncertain input parameters including their respective ranges has to be defined. Depending on the number of input parameters, a choice of input combinations of the parameters is selected within the parameter uncertainty space. As the emulator is required to predict the model output equally well across the $k$-dimensional parameter space, the input combinations have to be well-spaced and offer a good coverage. This is ensured by the use of maximin Latin hypercube sampling (Morris and Mitchell, 1995) to select these input combinations. We perform COSMO simulations for these input combinations and use them along with the corresponding outputs to train the emulators (*training data*). We used $15k$ input combinations to train the emulator, with $k$ the number of input parameters, which is 6 in S1, 7 in S2 and 6 in S3. Furthermore, 10 simulations were

added to the training datasets of S1 and S3 to increase the quality of the emulator fit. Thus, per Setup, 100 (S1 and S3) or 105 (S2) simulations were run to generate the training data.

The extension of a Gaussian distribution to an infinite number of variables is referred to as a Gaussian process (Rasmussen, 2004). A Gaussian process is defined by a mean function $m(\mathbf{x}) = \mathbf{h}(\mathbf{x})^T \beta$ and a covariance structure $V(\mathbf{x}, \mathbf{x}') = \sigma^2 c(\mathbf{x}, \mathbf{x}')$ where $\mathbf{x} = (x_1, \ldots, x_k)$ is a possible input combination, $\mathbf{h}(\mathbf{x})$ contains the regression coefficients for the mean functional form, $c(\mathbf{x}, \mathbf{x}')$ is a correlation function and $\beta$ and $\sigma^2$ are unknown coefficients. The specifications of the mean and the covariance reflect prior beliefs about the form of the emulator. We assume a linear trend for the mean function and use the Matérn correlation structure as it copes better with a slight roughness in the output surface (Rasmussen and Williams, 2006). These choices have been discussed in more detail by Lee et al. (2011), and have since then be used by a number of studies (Johnson et al., 2015; Igel et al., 2018; Wellmann et al., 2018; Glassmeier et al., 2019). Following the Bayesian paradigm, the a priori assumptions are updated using the training data by optimizing the marginal likelihood. The fitted emulator is then given by the resulting posterior specification of the Gaussian process (O'Hagan, 2004, 2006). Once an emulator is constructed, it needs to be validated to ensure an accurate estimation of the model output (Bastos and O'Hagan, 2009). For this, an additional 45 simulations with other input parameter combinations were conducted per setup. When comparing the emulator results to the results of the validation simulations, only a small number of outliers (up to 3) outside the 95% confidence intervals are accepted. In addition, a test for robustness of the choice of the training dataset has been conducted by interchanging the training dataset with parts of the validation data. The validated emulator is then able to predict (with a certain error as constrained by the validation) the output at all points in the multi-dimensional parameter uncertainty space that were not included in the training set and thus replaces the costly simulations of the NWP model.

## 3.2   Variance-based sensitivity analysis

Variance-based sensitivity analysis aims to decompose output variance into contributions from the uncertain input parameters. These include both contributions from each individual parameter and contributions from interactions of the parameters. The decomposition of the variance $V$ can be written as (Oakley and O'Hagan, 2004)

$$V = \sum_i V_i + \sum_{i<j} V_{ij} + \ldots + V_{1\ldots k} \tag{3}$$

assuming independence between the input parameters. $V_i$ are the individual contributions from each parameter, $V_{ij}$ denotes the contribution with respect to the interaction of two parameters, $i$ and $j$, up to $V_{1\ldots k}$ describing the joint interaction of all parameters together. To accomplish this decomposition, we use the extended Fourier amplitude sensitivity test (FAST) by Saltelli et al. (1999) where the k-dimensional parameter space is transformed to 1D Fourier space. Thus, the whole parameter space can be sampled by a monodimensional curve in the Fourier space. However, as several thousand runs would be necessary to get a space-filling curve, emulators are crucial for the required model output (Oakley and O'Hagan, 2004). A measure for the contribution from each parameter to the output uncertainty is given by the so-called *main effect* $S_i = \frac{V_i}{V}$, which we obtain by normalizing the variance contribution of the parameter $V_i$ with the overall variance $V$ in the output. Thus, the output variance could be reduced by the percentage given by $S_i$ if there was no uncertainty in the input $i$. Consequently, the difference between

the overall variance and the sum of the contributions of the individual parameters describes the amount of variance that arises from interactions of the parameters (*interaction effect*).

## 4   Sensitivity Analysis for variations of the microphysics (S2)

In the analysis, we consider several output variables for which emulators are derived as described above. These output variables, including vertically integrated hydrometeor mass contents, precipitation, diabatic heating rates and the size distribution of surface hail, will be described in more detail in this section. The results of the sensitivity analysis are shown for variations of the microphysical parameters only (S2). Similar analyses for variations of the environmental conditions (S1) have been discussed in Wellmann et al. (2018).

### 4.1   Hydrometeor mass contents and precipitation

The output variables of the model have to be reduced to 0 dimensions in order to be represented by the emulators. We are interested in the variables that are linked to severe weather at the surface (as precipitation maxima and hail), but also in the in-cloud processes causing them, and therefore in the microphysical properties of the cloud. To reduce the dimensionality of the output, the composition of the cloud is described by the vertically integrated mass content of each hydrometeor class that includes cloud water, hail, ice, snow, graupel and rain. The spatial and temporal mean is taken for the considered vertically integrated hydrometeor mass contents (all in kg m$^{-2}$).

The set of considered precipitation variables include the amount of hail at the ground per output interval of 15 minutes, the precipitation rate of hail and the total precipitation rate (all in kg m$^{-2}$ s$^{-1}$) and the accumulated total precipitation (in kg m$^{-2}$). Precipitation is analyzed similarly to the hydrometeor mass contents, but maximum values in space and time are considered instead of mean values. An exception is the amount of hail at the ground, for which both mean and maximum values are analyzed.

The results of the variance-based sensitivity analysis are shown as a bar plot in Fig. 1, where the hydrometeor mass contents are depicted on the left hand side and precipitation on the right hand side. Each bar represents one output variable, and the different colors denote the contributions from the input parameters to the output uncertainty (*Main Effect*). If there is blank space above the bar, this means that the first-order main effects are not able to explain all of the output uncertainty and that there are contributions from interactions of the input parameters.

Fig. 1 reveals that of the investigated parameters, the graupel fall velocity factor $a_G$ is the largest contributor to the output uncertainties of most of the integrated hydrometeor mass contents. For example, the uncertainty of the integrated cloud water content could be reduced by $43\%$ and the uncertainty of the integrated graupel content could even be reduced by $88\%$, if $a_G$ was known exactly. The second most important parameter is the CCN concentration, which contributes especially to the uncertainties of cloud water (in the microphysics scheme used here, primarily via an impact on autoconversion and thus on the partitioning between cloud and rain water) and snow content. In contrast, neither $a_G$ nor the CCN concentration are the

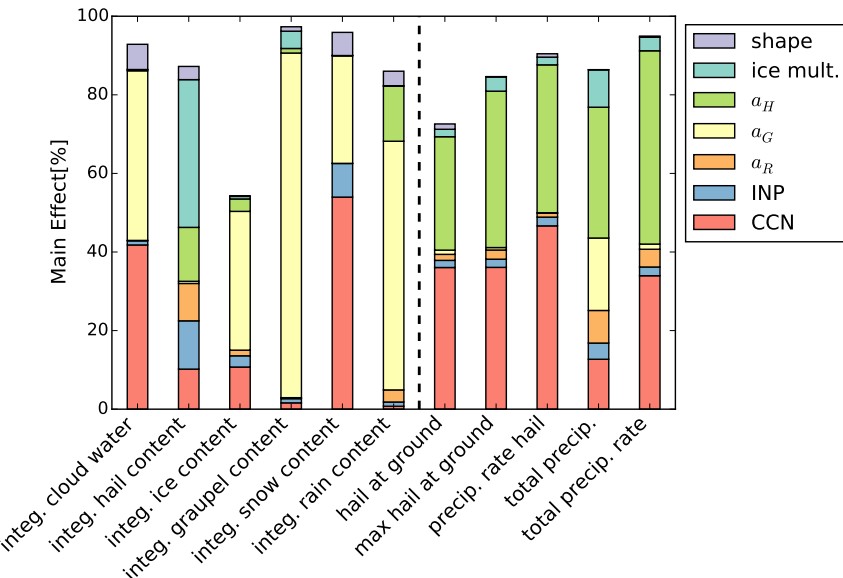

**Figure 1.** Bar plot of the main effect for vertically integrated hydrometeor mass contents (left) and precipitation (right) of cloudy grid points when only microphysical parameters are varied.

dominant parameters regarding the integrated hail content. Instead the strength of the ice multiplication is the largest contributor for that output variable (38% of the output uncertainty).

The output uncertainties of the considered precipitation variables are all dominated by contributions from the CCN concentration (13% − 47%) and the fall velocity of hail, modified by the scaling factor $a_H$ (29% − 49%). For the maximum total precipitation, the scaling factor for the fall speed of graupel, $a_G$, is also relevant. This is in line with the expectation that for for cases of strong convection, cold phase processes (including riming onto graupel) dominate precipitation formation, as was shown e.g. by Schneider et al. (2019).

## 4.2 Heating rates

Deep convective clouds usually cover a large area and thus are able to influence the surrounding atmosphere. Furthermore, diabatic processes cause a redistribution of energy such as heating due to condensation and freezing or cooling due to evaporation and melting. To examine how the simulated storm impacts the temperature profile, we interpret the vertical profiles of the diabatic heating rates. Joos and Wernli (2011) separate the associated temperature changes into contributions from phase transitions between the different hydrometeors such that it can be described as

$$\frac{\partial T}{\partial t} = \frac{L_v}{c_p}\left(S_C + S_R\right) + \frac{L_s}{c_p}\left(S_I + S_G + S_H + S_S\right) \tag{4}$$

where $L_v$ and $L_s$ are the latent heat of vaporization and sublimation and $c_p$ is the specific heat capacity of dry air for isobaric processes. The terms $S_x$ specify the conversion processes producing cloud water ($C$), rain ($R$), ice ($I$), graupel ($G$), hail ($H$)

or snow ($S$) that include phase transitions and therefore either supply or subtract energy from the surrounding air. Thus, the heating rate $\frac{\partial T}{\partial t}\big|_x$ related to each hydrometeor class $x$ is defined as

$$\frac{\partial T}{\partial t}\bigg|_x = \frac{L_{v,s}}{c_p} \cdot S_x \tag{5}$$

where $L_v$ is chosen for transitions between vapor and liquid, $L_s$ for transitions between vapor and ice and $L_s - L_v$ for transitions between liquid and ice. The spatial mean of the heating rates is calculated for each particle class in each layer. The temporal means of these profiles are predicted using separate emulators for each vertical level.

In order to obtain statistically robust results and to minimize the effect of single extreme events, emulators are used to generate $10,000$ realizations of the vertical profiles of the heating rates covering the whole parameter space. Subsequently, mean and standard deviation are calculated over all profiles together. Using this method, we are able to link changes of the total heating rate to the individual hydrometeor classes. Furthermore, the standard deviation is a measure of how much the heating rates react to variations of the input parameters. Fig. 2 shows the domain mean vertical profiles of the heating rates (left), where the shadings denote the standard deviation, and the corresponding main effects for the total heating rate in the considered vertical levels (right). Simulations with a near-identical model setup were analyzed by Barrett et al. (2019), and we refer to the hydrometeor profiles shown in their Fig. 3 and their process rate analysis for the rain water budget to support the interpretation of our results.

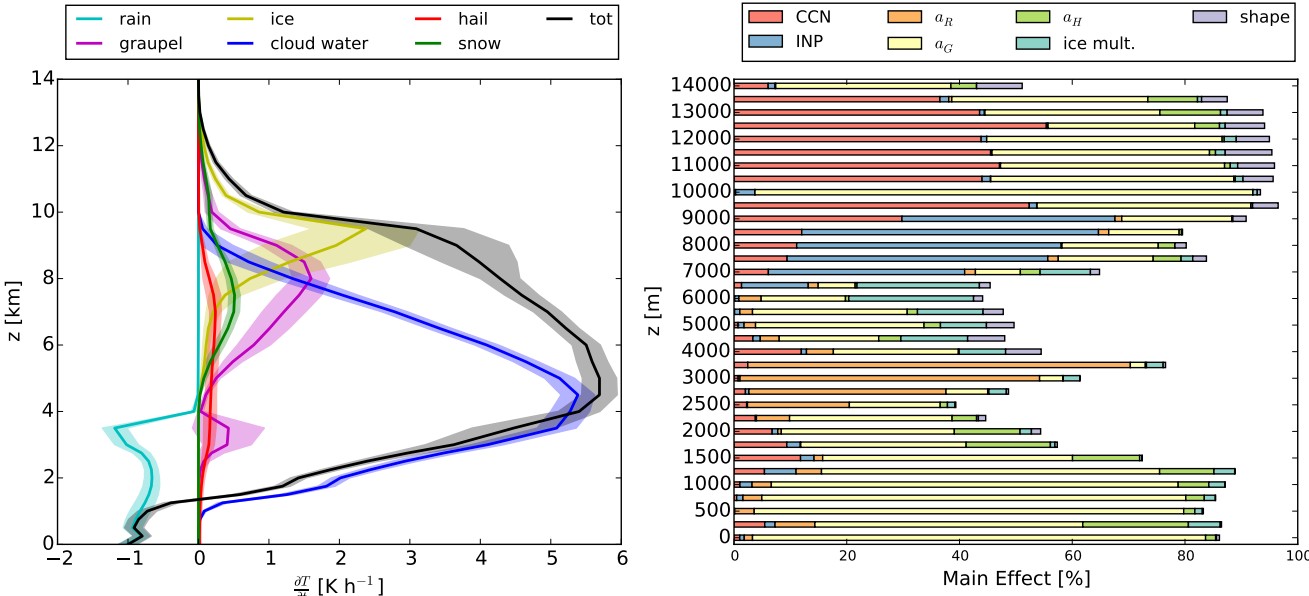

**Figure 2.** Left: Vertical profiles of the mean diabatic heating rates by each hydrometeor class and the mean total diabatic heating rate for variations of the microphysics. The shaded areas denote the standard deviation. Right: Bar plot of the corresponding main effect for the total heating rate. Note the different axis tick spacing below and above $3\,\mathrm{km}$.

Close to the ground the total heating rate is negative because of the cooling caused by evaporation of rain. As there is a strong increase of the heating due to the formation of cloud water, the total heating rate becomes positive above a height of about $1.3$ km and reaches its maximum of $5.7$ K h$^{-1}$ at $z = 5$ km. At higher altitudes, there are additional positive contributions from the formation of graupel and ice. However those are smaller than the contribution from the cloud water such that the

5 total heating rate decreases and is less than $1$ K h$^{-1}$ above $10$ km. In general, the profiles are quite robust to variations of the input parameters as the standard deviation is rather small (max. $20\%$ and on average less than $5\%$ of the absolute value for the total heating rate). The bar plot of the main effect (Fig. 2, right) reveals that the fall velocity of graupel ($a_G$) is the most important contributor to the output uncertainty of the total heating rate. In the height between $3$ km and $4$ km there are also major contributions from the fall velocity of rain ($a_R$). CCN concentration contributes only modestly to uncertainty at

10 these levels, although the heating rate by condensation is very strong here. This is probably linked to the fact that a saturation adjustment scheme is used for water vapor condensation, which is thus insensitive to droplet number and size. Below $2.5$ km, coinciding with the largest cooling due to the evaporation of rain, $a_G$ is again the major driver of uncertainty. As shown by Barrett et al. (2019), roughly half of the surface rain in this model setup originates from cold rain processes involving riming. Therefore here the graupel (and also hail) fall speed parameters contribute substantially to the uncertainty of the latent heating

rate at levels below $2$ km, although there is no graupel present at these altitudes.

Corresponding to the heating by the formation of ice between $7$ km and $10$ km, there are large contributions to the output uncertainty from the INP concentration in this height. Above, the output uncertainty of the total heating rate is dominated by the CCN concentration and the fall velocity of graupel. This is probably linked to the indirect effect of CCN and riming efficiency on the amount of supercooled water transported to the homogeneous freezing level. Furthermore, graupel is produced at these

20 levels in our model as a result of the freezing of rain drops, and the graupel fall speed factor thus impacts the gravitational sink of the (small) graupel particles present at these altitudes.

## 4.3 Size distribution of surface hail

The size distribution of hailstones reaching the ground is of interest regarding the damage potential of hail events. For the size distributions of hydrometeors, a generalized $\Gamma$-distribution is implemented in the two-moment scheme of Seifert and Beheng

(2006):

$$\frac{dN}{dx} = A x^\nu \exp\left(-\lambda x^\mu\right) \tag{6}$$

where $N$ is the number concentration, $x$ represents the particle mass and $\nu$ and $\mu$ are parameters of the $\Gamma$-distribution (cf. section 2.2). The coefficients $A$ and $\lambda$ are given by gamma distributions and the number and mass concentration, respectively (Seifert and Beheng, 2006). To obtain a measure for the number of particles per diameter, the term $\frac{dN}{dx}$ is transformed to

30 $\frac{dN}{dD}$ by a conversion from mass $x$ to particle diameter $D$. The spatio-temporal mean of the size distribution of surface hail is represented by emulators of the number concentration at ten fixed diameters. To constrain the parameter space and thus limiting the regimes describing different environmental or microphysical conditions to a feasible amount, each of the uncertain input parameters is assigned two discrete values where both a lower and a higher value are chosen (Table 3). These two values are

denoted by "-" and "+". Hereby, the outer bounds of the environmental parameters $F_{shear}$ and $\theta_0$ from S1 are taken as "-" and "+", as they are already limited to the typical range of forecast errors. For all other parameters, the lower and higher values are subjectively chosen to be representative, but not extreme, and encompass therefore a smaller range than examined in S1, S2 and S3. The considered regimes emerge from all possible combinations of these parameter values.

**Table 3.** Input values representing both lower and higher values of the parameter ranges used to analyze the size distribution of hail. Parameters marked with [*] are part of setup 1, [°] relates to setup 2 and [†] to setup 3.

| input | lower value (-) | higher value (+) | units |
|---|---|---|---|
| CCN concentration [*°†] | 500 | 3000 | $\text{cm}^{-3}$ |
| IN concentration [*°†] | 0.1 | 10 | scaling factor |
| wind shear ($F_{shear}$) [*†] | 0.3333 | 0.6666 | scaling factor |
| potential temperature $\theta_0$ [*†] | 299 | 301 | K |
| temperature perturbation $\Delta T$ (WB) [*] | 2 | 5 | K |
| radius of warm bubble $R_{hor}$ [*] | 7 | 13 | km |
| fall velocity of rain $a_R$ [°] | 0.5 | 1.5 | scaling factor |
| fall velocity of graupel $a_G$ [°†] | 0.5 | 1.5 | scaling factor |
| fall velocity of hail $a_H$ [°†] | 0.8 | 1.2 | scaling factor |
| ice multiplication [°] | $0.7 \cdot 10^8$ | $6.3 \cdot 10^8$ | $\text{kg}^{-1}$ |
| shape parameter [°] | 2 | 6 | - |

The size distribution of surface hail is simulated using the emulators for all possible combinations of the high and low input parameter values for each setup (128 combinations in S2, 64 combinations in S1 and S3). The aim of this approach is to attribute the minimum and maximum hail size distributions to specific parameter combinations. Fig. 3 (left) shows the mean size distributions of surface hail from all combinations and the corresponding main effect for variations of the microphysics only using S2. The size distributions with the lowest and highest number concentrations are marked in a different color such that a separation into three groups is visible.

The distributions in the two groups with either very low or very high number concentrations share common features regarding the combination of the input parameters. The lowest number concentrations of hail (over the entire size distribution) are found for regimes with a low value of the fall velocity of hail and a high value for the strength of the ice multiplication. These distributions show maximum number concentrations of $0.06 - 0.15\,\text{mm}^{-1}\,\text{m}^{-3}$ at a diameter of 7.5 mm. In contrast, the highest concentrations of $6.38\,\text{mm}^{-1}\,\text{m}^{-3}$ at a diameter of 5 mm are simulated for a high value of the fall velocity of hail. Thus, the fall velocity of hail and the strength of the ice multiplication are the most important controlling parameters of the size distribution.

The corresponding plot of the main effect (Fig. 3, right) confirms the impact of the fall velocity of hail ($a_H$) and the strength of the ice multiplication together to be responsible for large parts of the output uncertainty of the number concentration at

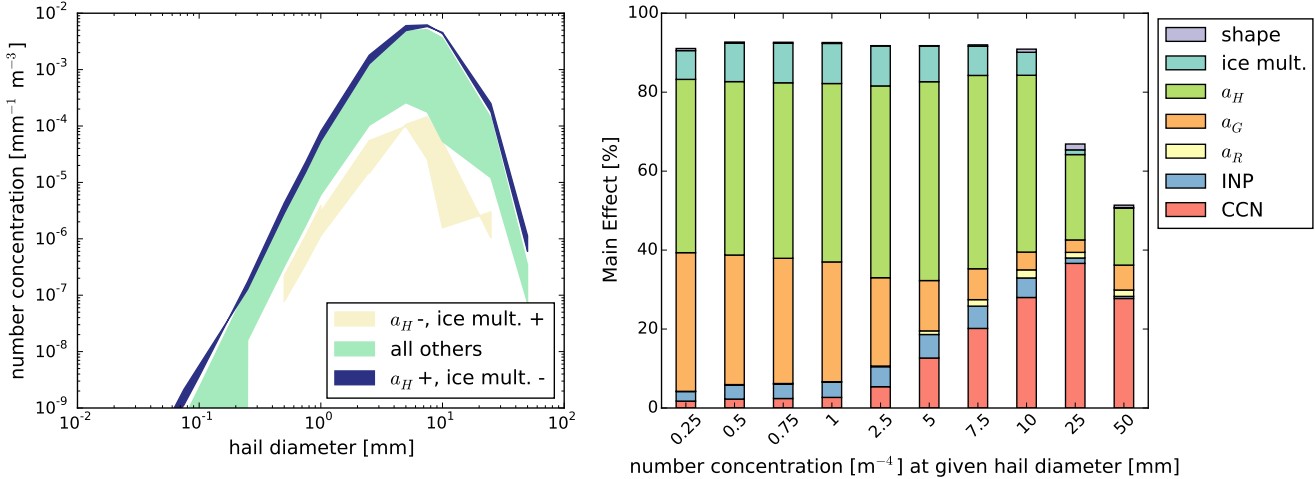

**Figure 3.** Left: Size distributions of hail at $z = 0$ m for variations of the microphysics. The shading illustrates regimes of the size distributions controlled by the fall velocity of hail. Right: Bar plot of the corresponding main effect for the number concentration of the size distribution of surface hail. Here, hail is defined according to the hydrometeor class in COSMO.

all considered diameters except at $D \leq 25$ mm. These two parameters contribute more than $50\%$ to the output uncertainty for these diameters. At the largest considered diameters, an increased contribution from the CCN concentration comes into play, while smaller diameters are significantly impacted by the graupel fall speed. This may be linked to the two formation pathways of hail in COSMO, namely through freezing of rain (of which the size is impacted by the CCN concentration) and through riming of graupel. A strong CCN impact on large hail particles was also found in two previous case studies (Loftus and Cotton, 2014; Khain et al., 2011), and related to CCN impacts on hail embryo sizes and the availability of supercooled liquid water for riming.

## 5 Comparison of the three setups

In the next step we analyze the impact of the input parameters on the uncertainty of the output variables of hydrometeor mass contents and precipitation by comparing the results for the three different setups with changes of 1) environmental conditions only, 2) microphysical parameters only and 3) both environmental conditions and microphysical parameters (S1 - S3, see sections 2.1-2.3). If the results of S3 resemble more those of S1, then the impact of the parameters describing the environmental conditions is more dominant. Correspondingly, the microphysical parameters are more dominant if S3 resembles S2.

## 5.1 Hydrometeor mass contents and precipitation

To compare the main effects of the three emulator studies, the results are combined in a bubble plot (Fig. 4) where the contribution of each considered input parameter to the output uncertainty is represented by the size of a circle. The circles of the different sets of input parameters are placed in columns next to each other labeled by *S1*, *S2* and *S3*.

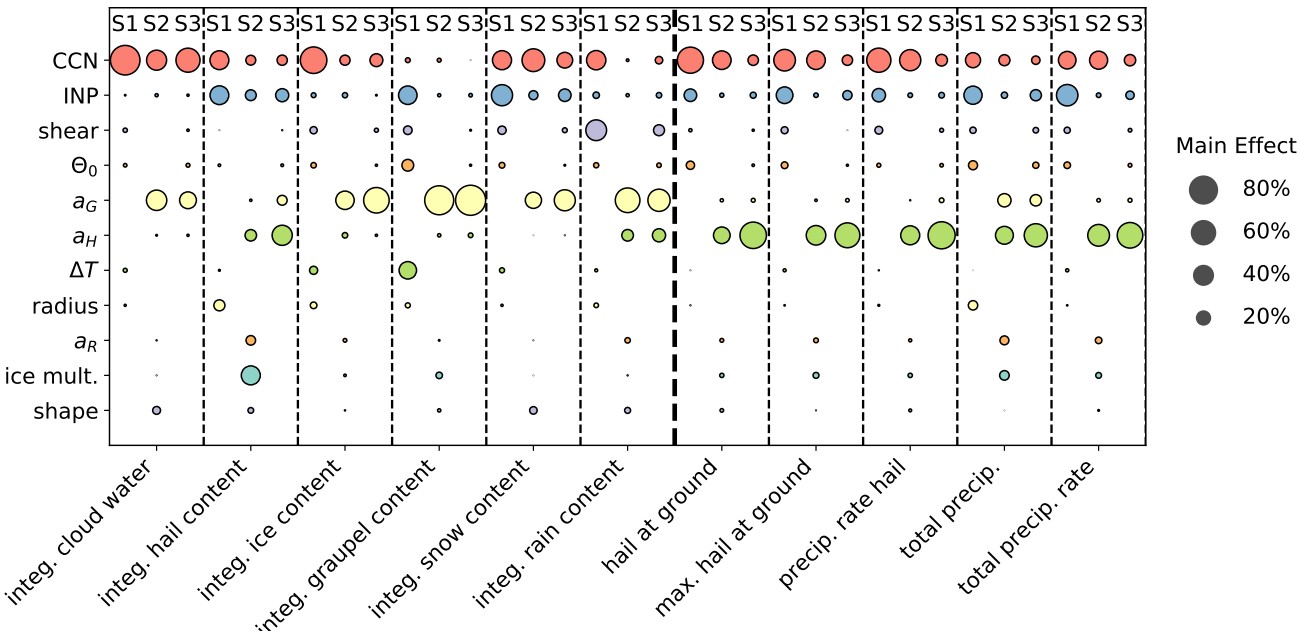

**Figure 4.** Bubble chart of the contributions from all input parameters of the different emulator studies to the output uncertainty of cloud and precipitation variables. The main effects of all input parameters given on the y-axis are depicted as circles where the size corresponds to the value of the main effect. The different columns labeled with S1, S2 and S3 represent the results of each emulator study (S1: environmental conditions, S2: microphysics, S3: both environmental conditions and microphysics; see sections 2.1-2.3). The numerical values for this figure are listed in Tables A1 and A2.

The CCN (100 to $4000 \, \text{cm}^{-3}$) and INP (factor $0.01$ to $10$) concentrations are changed within the same range in all setups such that the results from three separate ensembles can be compared. The contributions from the CCN concentration variations to the output uncertainty of the integrated cloud water and the integrated snow content in S3 are similar to those in S1. For the other variables, the contribution in S3 is rather comparable to the contribution in S2, while the contribution in S1 is larger. This trend is also consistent for the precipitation output. Here, the contribution from the CCN concentration uncertainty decreases from S1 to S3 such that the results of S3 are closer to those of S2.

The contributions from the INP concentration variations are mostly larger in S1 than in S2 for both integrated hydrometeor mass contents and precipitation. The main effects in S3 are a combination of S1 and S2, but the results are closer to those of S2 than to those of S1. Thus, the main effect of the INP concentration is smaller if other microphysical parameters are used as

input, possibly because other ice phase processes (secondary ice formation, riming) can suppress the sensitivity of a cloud to primary ice formation.

The behavior of the wind shear is quite consistent for the considered output variables. Its contribution is in general small, except if the intergrated rain water content is the target output variable. It is always larger in S1 than in S3, meaning that the wind shear has a larger impact on the output uncertainty, if only the environmental conditions are varied. Similarly, the (already small) impact of $\theta_0$ is reduced in S3; compared to the effect of cloud microphysics its impact is diminished.

The main effect of the fall velocity of graupel is larger for the cloud variables than for precipitation. Furthermore, in most of the cases the fall velocity of graupel has a similar effect on the output uncertainty in S3, such that $a_G$ is still important in cases when parameters describing the environmental conditions are also part of the input parameters.

When looking at the hydrometeor mass contents, the contribution from the fall velocity of hail to the output uncertainty is negligible except for the integrated hail and rain contents. However, it is the largest contributor to the uncertainty of the precipitation variables, presumably reflecting that hail itself and melted hail constitute a major part of the total precipitation. Here, its impact is larger in S3 compared to S2 for all variables so that its importance expands when also environmental conditions are involved.

The other input parameters ($\Delta T$, radius, $a_R$, the ice multiplication factor and the shape parameter) are only used in one of the setups so that a direct comparison of different setups is not possible. They are included in Fig. 4 for completeness.

Summarizing, we find that the uncertainty of the integrated hydrometeor mass contents and the precipitation mainly emerges from the uncertainty of the microphysics, in particular from the fall velocity of graupel for the hydrometeor mass contents and from the fall velocity of hail for precipitation. The contributions from the parameters characterizing the environmental conditions are rather small in S3.

In the literature, the focus of sensitivity studies is mainly on the effect of CCN concentrations on clouds, but there are also studies examining the effect of other parameters such as wind shear, temperature perturbation or shape parameter of the cloud droplet size distribution. For example, Brooks (1992) analyses the effect of the warm bubble characteristics on deep convection. He finds that variations of $\Delta T$ cause only minor differences in precipitation, and the updrafts are strongest for medium horizontal radii of the bubble. The effect of the horizontal radius on the precipitation is not mentioned. Our results are in good agreement with the findings of this work. Both $\Delta T$ and the radius of the bubble hardly contribute to the output uncertainty of the precipitation variables, and also the impact on the hydrometeor mass contents is rather small (Fig. 4). Regarding vertical wind shear, Dennis and Kumjian (2017) observe a significant effect of the wind shear on the hail production. Here, the contribution of the wind shear to the output uncertainties of hail variables is rather small. However, it is expected to see a larger impact when the wind shear does not have to compete with the more dominant effects of other parameters. Furthermore, in our study the parameter range of the wind shear is chosen to reflect typical forecast errors and not a broad range of atmospheric conditions. This results in a smaller impact of the wind shear variation compared to the setup of Dennis and Kumjian (2017).

The impact of CAPE on deep convection is analyzed by Storer et al. (2010). In their study, the updraft strength and the total accumulated precipitation are very sensitive to changes in CAPE, while the integrated amount of cloud water does not depend

strongly on CAPE. Furthermore, they conclude that the impacts of CAPE and CCN concentration can be comparable. Fig. 4 confirms that the contribution from $\theta_0$ to the uncertainty of the integrated cloud water in S1 and S3 is not dominant. Yet, in total the effect of the two parameters is not similar as the contributions from the CCN concentration are clearly larger. This is caused by the chosen parameter range of $\theta_0$ limited to typical forecast errors and thus not comparable to the parameter range assumed

by Storer et al. (2010). Igel and van den Heever (2017b) examine shallow cumulus clouds for different shape parameters of the cloud droplet size distribution and notice an effect on the droplet concentration, but not on the mass mixing ratios. The results of our study agree with their work, as the shape parameter is only of minor importance for the integrated cloud variables. With respect to the impact of CCN variation, our findings are in qualitative agreement with the works of Fan et al. (2013) and Yang et al. (2017), for instance. Fan et al. (2013) find an increase of approximately $30\%$ of the upper tropospheric cloud cover due

to changes of the CCN concentration from 280 to $1680\,\mathrm{cm}^{-3}$ (which is smaller than our parameter range). Yang et al. (2017) find clear differences in the vertically integrated condensate mixing ratio, such as an increase of ice from 6 to $18\,\mathrm{g\,kg}^{-1}$, for increasing CCN from 300 to $5000\,\mathrm{cm}^{-3}$ (similar to our parameter range). This is comparable to the significant influence of the CCN concentration on the output uncertainty of the hydrometeor mass contents found here.

## 5.2  Heating rates

In this study, the diagnostics of diabatic heating rates are implemented similar to Joos and Wernli (2011) (see section 4.2). The mean profile and the standard deviation of 10,000 randomly generated realizations are illustrated in Fig. 5.

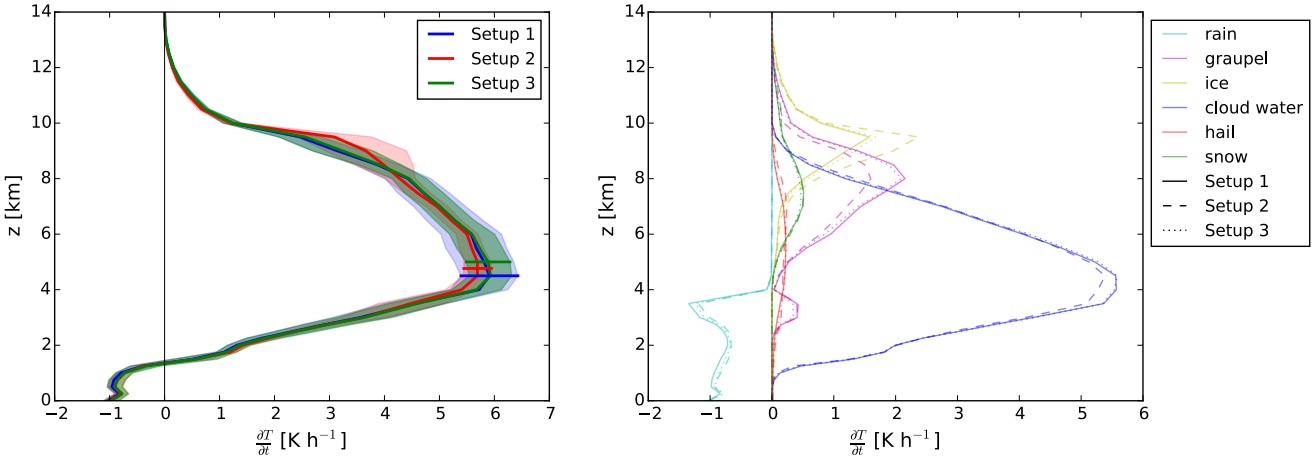

**Figure 5.** Vertical profiles of the mean total diabatic heating rate (left) and the mean heating rates for each hydrometeor class (right). The shaded areas left denote the standard deviation, which is also indicated by a horizontal bar at one selected altitude.

There is diabatic cooling of about $-1\,\mathrm{K\,h}^{-1}$ near the ground in all setups due to the evaporation of rain. Between $1.25$ and $1.5\,\mathrm{km}$ height the rate becomes positive and increases until its maximum is reached at a height of $4.5\,\mathrm{km}$. The maximum values of the heating rate vary between $5.7\,\mathrm{K\,h}^{-1}$ for setup 2 and $5.9\,\mathrm{K\,h}^{-1}$ for S1 and S3. Above, the total heating rate decreases

slowly up to $8$ km. Between $8$ and $10$ km there is a stronger decrease of the heating rate such that its value is close to $0$ K h$^{-1}$ at higher altitudes.

Up to $4$ km above the ground, the profiles of the mean heating rates are almost identical for the three considered setups. Also the standard deviations are small and almost negligible which means that near the ground the total heating rate is rather insensitive to changes of the input parameters, both environmental conditions and microphysical parameters. However, above $4$ km the profiles of S1 and S2 deviate from each other. The maximum of the total heating rate reached in S1 is slightly higher and the standard deviation enlarges to approximately $1$ K h$^{-1}$ while the standard deviation of S2 remains at values of $0.5$ K h$^{-1}$. The difference of the mean profile can be attributed to different contributions from the formation of cloud water that is smaller in S2 (Fig. 5, right). Here, the profile of S3 shows higher values and thus resembles the profile of S1. Another slight deviation of the profiles of S1 and S2 occurs in a height of $8-10$ km. At this point, the profile of S2 shows values that are up to $0.6$ K h$^{-1}$ larger than those of S1. Moreover, the standard deviation of S2 is increased to $0.7$ K h$^{-1}$ at these altitudes. This increase of the total heating rate in S2 is caused by an enhanced contribution from the formation of cloud ice at these altitudes as can be seen in Fig. 5, right. At this height, the profile of S3 is almost identical to that of S1. Above $10$ km the heating rates of all setups are close to each other showing only limited effects of the variations of the input parameters. Furthermore, the standard deviation of the profile of S3 is comparable to the standard deviation of S1, yet it is reduced by about $0.2$ K h$^{-1}$ in the middle troposphere. Therefore, variations of the environmental conditions have a larger impact on the total heating rate than variations of microphysical parameters. This dominance of the environmental conditions is also obvious in Fig. 5 (right). Near the ground, the total heating rate is determined by the cooling due to evaporation of rain, while in the mid-troposphere the largest contributions stems from the formation of cloud water mainly caused by the use of saturation adjustment in the microphysics scheme. At higher altitudes the hydrometeors of the ice phase, especially graupel and cloud ice, contribute the most to the total heating rate. For all hydrometeors, the profiles of S3 (dotted) are close to those of S1 (solid), whereas the profiles of S2 (dashed) differ. Thus, the environmental conditions dominate the impact on the vertical profiles of the heating rates for both the total heating rate and the individual heating rate contributions from each hydrometeor class.

Condensation of cloud water, which is a substantial contributor to the total heating rate in the lower and middle troposphere, is parameterized via a saturation adjustment scheme in our model. Nevertheless, it yields a large contribution to output uncertainty of the diabatic heating in all three setups. This effect might be even larger if a time-dependent treatment of condensation was used. Wang et al. (2013), for example, find that there are discrepancies of the results between models including saturation adjustment and those explicitly calculating diffusional growth of cloud droplets. These differences are mainly characterized by an overestimation of the condensation in the lower troposphere affecting the diabatic heating rates. In addition, Lebo et al. (2012) also state that saturation adjustment artificially increases condensation. This increase appears to be quite strong as it is also represented by the emulators. Therefore, modified results of the sensitivity studies are expected for the heating rates, if the saturation adjustment is replaced by more realistic calculations.

## 5.3 Size distribution of surface hail

In this section, we analyze the impact of variations of environmental conditions and microphysical parameters on the size distribution of surface hail. As described in section 4.3, each input parameter is assigned two discrete values and the size distribution is predicted by the emulators for all possible combinations. In Fig. 6 both the distributions with the lowest and highest number concentrations are illustrated for each setup. Consequently, all other distributions are found in-between which is indicated by the shading. The combinations of the parameters producing the extreme distributions, and thus the controlling input parameters of the size distributions, are given in the legend.

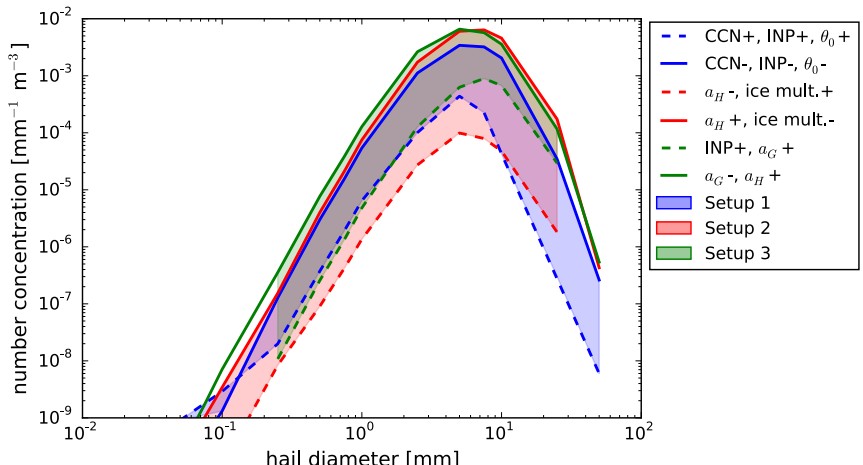

**Figure 6.** Size distributions of hail at $z = 0$ m. The shading illustrates the number concentrations covered by all possible combinations of input parameters for each setup. The solid lines indicate the distributions with the highest number concentration while the dashed line represents the distributions with the lowest number concentration of each setup. The corresponding combination of controlling input parameters is given in the legend.

For S1, the size distribution with the lowest number concentration (dashed blue line) has its maximum of $4 \times 10^{-4}$ mm$^{-1}$ m$^{-3}$ at a hail diameter of $5$ mm. The maximum of the distribution with the highest number concentration (continuous blue line) is also found at the same diameter but with a number concentration of $3.4 \times 10^{-3}$ mm$^{-1}$ m$^{-3}$. For this setup (in which the environmental conditions are modified), the controlling parameters are the CCN and INP concentrations and $\theta_0$. Low number concentrations of hail arise for higher values of these parameters and high number concentrations of hail for lower values.

The maximum of the size distribution with low number concentrations of S2 (dashed red line) is only a fourth of the concentration of S1 while for the distributions with the highest number concentration (continuous red line) it is almost twice the amount. Hence, the spread of all distributions is larger.

For S2, the low (dashed red line) and high (continuous red line) hail size distributions are smaller and larger, respectively, than those for S1, leading to a larger spread in the distributions. The fall velocity of hail and the strength of the ice multiplication are the two microphysical parameters that mainly determine the number concentration of surface hail. Low number concentrations

are found for a low value of the fall velocity of hail combined with a high value for the strength of the ice multiplication and vice versa.

When both the environmental conditions and the microphysics are perturbed, the lower limit of the size-resolved number concentration of hailstones approximately doubles compared to S1. The distribution with the highest number concentration has similar concentrations as S2. The combination of high INP concentrations and high fall velocities of graupel produce a low number concentration of surface hail whereas low fall velocities of graupel (presumably resulting in more time for riming of graupel and growth to hail) and high fall velocities of hail (possibly by leaving less time for melting below the cloud) lead to high number concentrations.

Comparing the results of the different setups, the distribution with the lowest number concentration of S3 is similar to the corresponding distribution of S1. Especially for small diameters the two distributions show similar number concentrations. In contrast, the distribution with the highest number concentration of S3 (continuous green line) resembles the distribution of S2 as high number concentrations are reached that are comparable to S2. Furthermore, the spread between the distribution with the lowest and the highest number concentration is smaller in S1 and larger in S2 such that the spread of S3 is situated in-between. Moreover, the controlling parameters identified in S3 include parameters from both environmental conditions (INP) and microphysics ($a_G$, $a_H$).

Summarizing, the environmental conditions and the microphysical parameters (with the spread of input parameters chosen in this study) have a comparable impact on the size distribution of surface hail. While the microphysical input parameters mainly determine the maximum number concentration, the environmental conditions substantially influence the minimum number concentration. In general, microphysical input parameters cause a larger spread of the number concentrations of surface hail than the inputs related to environmental conditions.

The results above should not be regarded as definite number concentrations of surface hail, as a bulk model is used here, and several studies note that the representation of hydrometeor sizes is more accurate in bin schemes (Dennis and Kumjian, 2017; Lee et al., 2008). To approach this issue, Loftus and Cotton (2014) introduce a modified microphysics setup where a three-moment scheme is implemented for an improved prediction of hail. They find that increasing the CCN concentration induces an increase of the hail sizes, but a decrease of the number of hailstones. The CCN concentration is identified as the controlling parameter of the size distribution in this study as well, but not for all considered setups. Because Loftus and Cotton (2014) investigated the effect of the CCN concentration only, it is possible that in our study the effect of the CCN concentration is covered by larger impacts of other input parameters such as the fall velocity of hail. Thus, the classification of the controlling parameters of the size distribution of hail is assumed to be appropriate although a bulk microphysics scheme is used. Further studies similar to Loftus and Cotton (2014), incorporating modifications of the microphysics scheme and the variation of not only one but several parameters, are necessary to confirm these findings.

# 6    Summary & Conclusions

In our study, we have investigated how changes in the environmental conditions and cloud microphysics impact deep convection with a focus on the integrated hydrometeor mass contents, precipitation, diabatic heating rates and the hail size spectrum.

The COSMO model was used to simulate deep convective clouds in an idealized setup, where convection was triggered by an artificial warm bubble. This rather simple setup was required to allow a large number of simulations in which environmental conditions and microphysical parameters are modified. The straightforward approach for analyzing the sensitivity of the model output to changes in the input parameters is to vary a chosen parameter in a given range, while the other parameters are kept constant. However, instead of this one-at-a-time analysis, we employed statistical emulation and variance-based sensitivity analysis where the contributions of the input parameters to the uncertainty of the output are quantified. The emulator approach offers a convenient tool for the identification of relevant parameters without the requirement of running a large number of extensive model simulations. COSMO simulations were used to train the emulators, while the variance-based sensitivity was based on the predictions from the emulators allowing for an identification of not only the impact of each parameter independently, but also their interactions which cannot be captured by one-at-a-time analyses. In total, we evaluated three sets of input parameters. First, a set describing environmental conditions such as potential temperature and vertical wind shear was used. Note that the range of variation of these parameters is designed to mimic typical forecast errors and is therefore smaller than in earlier studies, which have encompassed a wider range of possible conditions. The second set of input parameters focused on cloud microphysics consisting of parameters such as the shape parameter of the cloud droplet size distribution or the fall velocity of hydrometeors. The third set combined influential parameters of both environmental conditions and microphysics. For all sets of input parameters, the integrated hydrometeor mass contents, precipitation, size distribution of surface hail and diabatic heating rates were examined with respect to output uncertainty or response to variations of the input.

The analysis of the integrated hydrometeor mass contents reveals that the CCN concentration is an important parameter contributing to the output uncertainty if only the environmental conditions are varied, whereas the fall velocity of graupel provides a large contribution if only microphysical parameters are varied. These parameters are crucial for the efficiency of warm and cold rain formation, respectively. The decomposition of the output variance given variations of both environmental and microphysical parameters is similar to variations of the microphysical parameters only, implying that regarding the integrated hydrometeor mass contents, the uncertainty in the microphysical parameters is more dominant in causing uncertainty in the output. Similarly, the CCN and INP concentrations are relevant parameters for the uncertainty of the precipitation output when environmental conditions are considered, while the CCN concentration and the fall velocity of hail dominate when microphysical parameters are analyzed. The study combining both sets of input parameters shows a large contribution by the fall velocity of graupel to the output uncertainty of the hydrometeor loads, and by the fall velocity of hail to the output uncertainty of the precipitation variables. Consequently, variations of the microphysical parameters are the prevailing source of uncertainty of the integrated hydrometeor mass contents and precipitation compared to variations of the environmental conditions.

We analyzed the variability of the vertical profiles of the diabatic heating rates by using emulators to predict the profiles of $10,000$ randomly generated realizations covering the whole parameter space. The mean profiles for the three sets are almost identical, with the exception of a deviation of the set with variations in microphysical parameters in the middle and upper troposphere. The variability is similar for the set with variations of environmental conditions only and the set with combined microphysical and environmental changes. The good agreement between the results of these two sets of input parameters is also confirmed by the component-wise analysis of the heating rates where the contribution from each hydrometeor class to the total heating rate is considered separately. Thus, comparing the impact of environmental conditions and the microphysics on the diabatic heating rates, the effect of the environmental conditions is dominant. This is in contrast to the result of the integrated hydrometeor mass contents and precipitation where the impact of the microphysical parameters is prevalent.

We have assigned two discrete values to each of the input parameters and then used the emulators to predict the hail size distribution for all possible combinations of the input parameters to understand how the surface hail is affected by variations of the environmental conditions and the microphysics. The parameters controlling the size distribution are the CCN concentration, the INP concentration and the vertical temperature profile for variations of the environmental conditions and the fall velocity of hail and the strength of the ice multiplication for variations of the microphysics. The controlling parameters of the combined input parameters are the INP concentration and the fall velocities of graupel and hail. The range of number concentrations in which the size distributions are found in this combined set is a compromise of the two sets considering environment and microphysics separately where the distribution with the lowest number concentration is close to the results for variations of the environmental conditions and the distribution with the highest number concentration is close to the results for variations of the model microphysics. Accordingly, both the environmental conditions and the microphysics affect the size distribution of surface hail comparably.

In conclusion, the aim of this work was to identify the sources of forecast uncertainty and to determine whether the variation of the environmental conditions or the variation of the microphysical parameters leads to larger model output uncertainty. It can be expected that our results (in particular regarding the microphysical parameters) depend to some extent on the microphysics scheme of our model. However, the overarching aim of this study was not to emphasize the impact of a specific parameter, but to quantify the relevance of environmental versus microphysical uncertainty in general. We expect that these results are less dependent on the microphysics scheme. In addition, future studies should address how far the results of our idealized simulations are transferable to real cases. For our choices of input parameter ranges, the impact of the environmental conditions versus cloud microphysics depends on the output of interest: The uncertainty in the output of the integrated hydrometeor mass contents and the precipitation is affected more by variations of the microphysics, while variations of the environmental conditions cause more uncertainty in the prediction of the vertical profiles of the diabatic heating rates. Further, a comparable impact of environmental conditions and microphysics on the size distribution of surface hail is found. Therefore, depending on the parameter of interest, the forecast uncertainty could be reduced by either an improved observational network

and data assimilation providing a more accurate description of the environmental conditions or a revised microphysics scheme, in particular a revised parameterization of the fall velocity of graupel and hail.

*Data availability.* The processed training data sets and the emulators are published via the open access institutional repository KITopen (doi:10.5445/IR/1000099232). The full model simulations are available upon request.

*Author contributions.* JJ and KC provided the code for the emulator approach. CW conducted the analysis and wrote the original draft of the paper with contributions from the co-authors. CH conceptualized the project together with MK and BV, and edited the revised manuscript. CH, AB, MK and BV contributed to the discussion and interpretation of the results.

*Competing interests.* The authors declare that they have no conflict of interest.

*Acknowledgements.* The research leading to these results has been done within the project "Microphysical uncertainties in deep convective
clouds and their implications for data assimilation" of the Transregional Collaborative Research Center SFB / TRR 165 "Waves to Weather" funded by the German Science Foundation (DFG). JJ and KC were supported by the Natural Environment Research Council ACID-PRUF project under grant NE/I020059/1. The simulations were performed on the computational resource ForHLR I funded by the Ministry of Science, Research and the Arts Baden-Württemberg and DFG. Furthermore, we thank Felix Fundel from Deutscher Wetterdienst for providing data on the prediction errors of the COSMO model.

**Appendix A: Numerical values**

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

**Table A1.** Numerical values represented by the circles in Fig. 4 for the integrated hydrometeor mass contents. All values are given in %.

| | integ. cloud water | | | integ. hail content | | | integ. ice content | | | integ. graupel content | | | integ. snow content | | | integ. rain content | | |
|---|---|---|---|---|---|---|---|---|---|---|---|---|---|---|---|---|---|---|
| | S1 | S2 | S3 | S1 | S2 | S3 | S1 | S2 | S3 | S1 | S2 | S3 | S1 | S2 | S3 | S1 | S2 | S3 |
| CCN | 91.2 | 41.8 | 60.3 | 37.2 | 10.2 | 11.0 | 73.4 | 10.7 | 17.3 | 2.4 | 1.6 | 0.006 | 37.9 | 54.0 | 25.8 | 38.9 | 0.7 | 5.9 |
| INP | 0.2 | 1.1 | 0.5 | 36.8 | 12.3 | 17.9 | 2.4 | 2.9 | 0.3 | 35.7 | 1.1 | 1.3 | 46.4 | 8.6 | 16.4 | 4.3 | 1.1 | 3.1 |
| shear | 2.0 | - | 0.6 | 0.01 | - | 0.09 | 5.4 | - | 1.7 | 7.7 | - | 0.3 | 7.2 | - | 2.5 | 44.9 | - | 12.5 |
| $\theta_0$ | 1.3 | - | 1.6 | 0.7 | - | 0.8 | 3.1 | - | 0.5 | 14.4 | - | 0.5 | 3.5 | - | 0.3 | 2.8 | - | 2.1 |
| $a_G$ | - | 43.2 | 29.4 | - | 0.5 | 10.1 | - | 35.3 | 67.9 | - | 87.7 | 94.3 | - | 27.4 | 46.1 | - | 63.3 | 51.4 |
| $a_H$ | - | 0.3 | 0.5 | - | 13.7 | 42.3 | - | 3.2 | 0.5 | - | 1.2 | 2.3 | - | 0.005 | 0.04 | - | 14.0 | 18.1 |
| $\Delta$T | 1.7 | - | - | 0.4 | - | - | 6.9 | - | - | 31.7 | - | - | 2.7 | - | - | 0.9 | - | - |
| radius | 0.5 | - | - | 12.7 | - | - | 4.6 | - | - | 2.6 | - | - | 0.5 | - | - | 2.2 | - | - |
| $a_R$ | - | 0.1 | - | - | 9.5 | - | - | 1.4 | - | - | 0.3 | - | - | 0.02 | - | - | 3.1 | - |
| ice mult. | - | 0.003 | - | - | 37.6 | - | - | 0.6 | - | - | 4.4 | - | - | 0.008 | - | - | 0.09 | - |
| shape | - | 6.4 | - | - | 3.4 | - | - | 0.1 | - | - | 1.1 | - | - | 5.8 | - | - | 3.7 | - |

**Table A2.** Numerical values represented by the circles in Fig. 4 for the precipitation variables. All values are given in %.

| | hail at ground | | | max. hail at ground | | | precip. rate hail | | | total precip. | | | total precip. rate | | |
|---|---|---|---|---|---|---|---|---|---|---|---|---|---|---|---|
| | S1 | S2 | S3 | S1 | S2 | S3 | S1 | S2 | S3 | S1 | S2 | S3 | S1 | S2 | S3 |
| CCN | 71.2 | 36.1 | 11.4 | 48.6 | 36.1 | 11.2 | 60.8 | 46.6 | 14.2 | 23.7 | 12.7 | 7.4 | 32.2 | 34.0 | 13.5 |
| INP | 16.5 | 1.8 | 3.8 | 29.0 | 2.1 | 9.0 | 18.8 | 2.2 | 3.4 | 34.5 | 4.1 | 13.0 | 47.9 | 2.2 | 7.3 |
| shear | 1.1 | - | 0.5 | 5.1 | - | 0.008 | 6.4 | - | 1.6 | 4.1 | - | 3.2 | 4.0 | - | 1.4 |
| $\theta_0$ | 7.2 | - | 0.7 | 4.8 | - | 0.5 | 1.7 | - | 1.3 | 8.5 | - | 4.0 | 4.5 | - | 1.5 |
| $a_G$ | - | 1.0 | 1.6 | - | 0.6 | 1.3 | - | 0.1 | 2.3 | - | 18.4 | 13.5 | - | 1.3 | 1.7 |
| $a_H$ | - | 28.9 | 73.0 | - | 39.8 | 64.5 | - | 37.5 | 76.5 | - | 33.3 | 56.0 | - | 49.2 | 68.2 |
| $\Delta T$ | 0.01 | - | - | 1.0 | - | - | 0.1 | - | - | 0.002 | - | - | 1.0 | - | - |
| radius | 0.1 | - | - | 0.2 | - | - | 0.4 | - | - | 9.2 | - | - | 0.2 | - | - |
| $a_R$ | - | 1.5 | - | - | 2.3 | - | - | 1.0 | - | - | 8.3 | - | - | 4.5 | - |
| ice mult. | - | 1.9 | - | - | 3.7 | - | - | 2.0 | - | - | 9.5 | - | - | 3.4 | - |
| shape | - | 1.4 | - | - | 0.05 | - | - | 0.9 | - | - | 0.005 | - | - | 0.3 | - |