# Peer review of "Comparing the impact of environmental conditions and microphysics on the forecast uncertainty of deep convective clouds and hail"

_Atmospheric Chemistry and Physics, 2019_

## Referee Comment (RC1) · Anonymous Referee #1 · 12 Aug 2019

Review of "Comparing the impact of environmental conditions and microphysics on the forecast uncertainty of deep convective clouds and hail" by Wellmann et al.

This study examines the relative influence of environmental conditions and microphysical parameters on vertically integrated hydrometeor contents and precipitation values (with an emphasis on hail), in addition to their influence on heating rates. The authors use an emulator technique to reduce the number of simulations that would otherwise be required to produce the presented results. In general, the study finds that environmental conditions and microphysical properties contribute to forecast uncertainty; however, when the environment and microphysics interact with each other, the latter

tends to dominate forecast uncertainty in hydrometeor contents and precipitation.

Overall, I have a long list of comments; most importantly, I think that the authors should provide more analysis and discussion in several of the sections (please see general and specific comments). The paper is well written, but readability could be improved by a more liberal use of commas in addition to the rewording of several sentences (please see specific comments). In addition, the authors should try to be consistent about their use of active versus passive voice. With all of this said, I think that the results are interesting and worthy of publication, and at this stage I suggest acceptance subject to major revision.

Major/general comments:

1. Model Setup (Section 2) and Methods (Section 3): For clarity, and especially for those readers who are not familiar with the emulator technique, there should probably be more information included about the modeling approach. For example, what is the total number of simulations conducted? Which "input combinations" are simulated? How does your choice for the mean function and correlation structure influence the results? How do you validate the emulator? Are the emulator results sensitive to the chosen minimum and maximum parameter values? What is the argument for including CCN and INP in the environmental conditions setup? Do the results change notably if these two microphysical properties are included only in the microphysical conditions setup?

2. Sensitivity Analysis for Variations of the Microphysics (S2) (Section 4): This section, which represents the bulk of the paper, generally lacks depth and therefore should contain additional insight and discussion. For instance: [P9, L14-20] Why look at the mean for hydrometeor content, max for precipitation, and both for amount of hail? [P9, L31-32] What about snow and hail as contributors to the output uncertainties? Maybe this should say one of the largest contributors. [P9, L39-40] Total precipitation, which is a very important quantity, seems to be affected more notably by the fall speed of

graupel scaling factor than by CCN. Please comment on this.

3. Heating rates (Section 4.2): The results from this section are quite interesting; however, I feel as though it is lacking a bit in terms of analysis and discussion. It would be nice if the authors took some time to dive a little deeper. For example: [P11, L17-18] Why is the fall speed of graupel the most important at low altitudes (<2 km) and high altitudes (>10 km), which is where the graupel heating rates are very small if not zero? Is graupel present in these regions? Perhaps a figure showing vertical profiles of hydrometeor contents may help. [P11, L18-19] Where the fall speed of rain plays a notable role in the main effect (between 3 and 4 km), the magnitude of the rain heating rate is only a small fraction of the magnitude of the total heating rate. Does this suggest that, in general, the model physics is more uncertain about rain evaporation processes than cloud condensation processes? [P12, L2-3] Can you speculate as to why the CCN concentration and fall speed of graupel dominate the total heating rate output uncertainty at high altitudes?

4. Hydrometeor masses and precipitation (Section 5.1): In general, this section would benefit from a deeper (and more quantitative) analysis. Figure 4 is really nice for visual comparison; however, can the numerical values be put into a table (perhaps in an appendix or a supplement) for a more quantitative comparison? Also, to minimize reader confusion, I recommend not putting a circle in areas where the input parameter was not part of the emulator simulation. For instance, under S2 for shear, under S1 for graupel fall speed, etc. Some comments and questions about the section text: [P14, L4-6] When referring to the trend for precipitation output, what about for total precipitation rate, which appears to be different? Is this important? Again, a table would help clarify these comparisons. [P15, L3] When referring to the main effect of the INP concentration, what is the physical interpretation? The influence of uncertainty in the INP concentration is muddled when the uncertainty in the other (individual? some? all?) microphysical parameters are introduced? Please elaborate.

Minor/specific comments:

1. P1, L15-17: Maybe reword to make more clear that you are emphasizing environmental parameters and microphysical parameters. Also, please separate the citations to better associate with these two different aspects of forecasting convective clouds.

2. P2, L3: What are the different choices of the trigger? 3. P2, L5-6: Please add references for the Morrison and Thompson schemes.

4. P2, L6-7: Which aspects of the parameterizations are most influential?

5. P2, L8: Individual parameters such as?

6. P4, L7: Horizontal resolution or grid spacing?

7. P4, L9: Can you provide approximate vertical grid spacings in the layer(s) of interest?

8. P4, L9: Do the open boundary conditions cause any mass conservation issues?

9. P4, L20: How are cloudy grid boxes defined?

10. P4, L26: Which "two former studies"?

11. P5, Table 1: The fourth input says "potential temperature at the ground", but the text says "vertical temperature profile". Please clarify.

12. P5, L6: Perhaps "maritime" should be changed to "clean" because marine cloud can be polluted.

13. P5, L17-18: This sentence is confusing...does it turn toward the west until a straight easterly flow is reached or does it turn toward the east until a straight westerly flow is reached?

14. P5, L20: Please state explicitly the wind direction bounds at the surface.

15. P5, L20: Please add a citation for this statement.

16. P5, L23: Please add a citation for this statement.

17. Table 2: Please add the symbol/abbreviation for the various parameter inputs (where necessary).

18. P7, L11: Why not also vary $\mu$?

19. P7, L32: Please change "data" to "output".

20. P8, L20-21: Can you comment on the errors that are associated with this prediction?

21. P9, L16: Contents or vertically integrated contents?

22. Figure 2: Is it possible to coordinate the y-axes of the two panels so that a direct comparison in the vertical is easier?

23. P11, L13-15: While this is true, perhaps note that the total heating rate does not decrease as rapidly as the rate due to cloud water because graupel and ice heating rates are at a maximum between about 8 and 10km.

24. P12, L31-33: The impact of the fall speed of hail and the strength of ice multiplication is mentioned as being important, but what about the role of the fall speed of graupel? This seems quite important for small hail. Moreover, can you speculate as to why CCN concentration becomes more important at larger hail diameters?

25. Table 3: What is the difference between assigning the input parameters "lower" and "higher" values as opposed to "min" and "max" values (as in Tables 1 and 2)? Also, why are the values used in this experiment different from those used for S1 and S2? Is the input listed as "potential temperature" at the surface?

26. P14, L1 (below Fig. 4): Can you parenthetically reference the CCN and INP concentrations here to help the reader? 27. P15, L4-7: Maybe note that, in general, uncertainties in wind shear (which is likely not uncommon in numerical weather prediction) do not have a notable impact on the output variables examined here with the exception of integrated rain water content (rain water path).

28. P15, L6: Perhaps note that the impact of theta is already relatively low.

29. P15, L31-32 ("The remaining input parameters [...]"): I am not sure that I understand this sentence. Please clarify.

30. P15, L34-35 ("in particular from the fall velocity of graupel for the hydrometeor masses and from the fall velocity of hail for precipitation"): This seems like an important finding; please italicize for emphasis.

31. P15, L35-36: So, are you able to say that uncertainties in the selected environmental conditions are muddled by uncertainties in the selected microphysical parameters?

32. P15, L50-51 ("In their study [...]"): This sentence is a bit confusing; please reword and/or flesh out.

33. P16, L7: To which aerosol effect(s) are you referring?

34. P16, L8-11: Can you be more specific about the Fan et al. (2013) results? Changes up to 25%? 25% on average? Is their range of CCN concentration similar to the one used here? Also, for the Yang et al. (2017) paper, what range of CCN concentration was tested? This is important when comparing previous results to results from the current study.

35. Figure 5: For the left panel, please mark the 0 K/h value to delineate between negative and positive values. Also, for both panels, is it possible to make the lines thicker in this figure (as in Fig. 2)?

36. P17, L17-18: When talking about the saturation adjustment in the microphysics scheme, how are you able to make this conclusion? Please elaborate.

37. P17, L33-34: When talking about the emulator-predicted size distributions, are you able to comment on the expected uncertainty in your results?

38. Figure 6: As in Fig. 5, are you able to make the lines thicker?

39. Size distribution of surface hail (Section 5.3): For the analysis in this section, please use line color and style to help clarify to which distribution you are referring.

40. P18, L5: When referring to the value of 0.4 mm-1 m-3, this number does not seem to correspond with the y-axis. Am I missing something here?

41. P18, L7: Similar to comment #40, when referring to the value of 3.4 mm-1 m-3, this number does not seem to correspond with the y-axis.

42. P18, L7-9: This sentence is confusing. Perhaps say something like: "when relatively high (low) values of theta and high (low) concentrations of CCN and INP are present, low (high) hail number concentrations result".

43. P18, L10-12: This could probably be stated more simply in one sentence. For example, something like: "For S2, the maximum and minimum of the hail size distribution is larger and smaller than that for S1, leading to a larger spread in the distributions."

44. P19, L3-10: What about for larger diameters? It is especially interesting that the number concentrations for the minimum size distributions are largest for S3. This suggests that, individually and for the lower bound, S1 and S2 do not produce large number concentrations, but if the environmental and microphysical conditions are combined (S3), then there is an enhancement. Please comment on this notable difference.

45. P19, L8-10: So, can you speculate as to what this means physically?

46. P19, L12-13 ("While the microphysical input parameters mainly determine the maximum number concentration, the environmental conditions substantially influence the minimum number concentration"): This seems like an important finding; please italicize for emphasis. 47. P20, L17-27: There is a large body of literature, some of which is referenced in the introduction, that focuses on the so-called "aerosol invigoration" hypothesis. Can you relate the work presented here to previous work?

48. P21, L19-20: What about also a revised parameterization of the fall speed of graupel?

Grammatical/wording recommendations:

1. The authors interchange hydrometeor "content" and "mass" throughout the text. Which parameter is actually shown? Please be consistent.

2. P2, L4-5: Awkward sentence; please reword.

3. P2, L5: Change "three cloud types for" to "three cloud types using".

4. P2, L19: Maybe "Additional relevant" instead of "Further relevant"?

5. P2, L28: Change "Moreover, field studies indicate that fall speeds of hydrometeors are observed in a broad range of velocities." to "Moreover, field study observations indicate that hydrometeors may have a broad range of fall velocities."

6. P2, L30-32: Awkward sentence; please reword.

7. P3, L10-12: Awkward sentence; please reword.

8. P3, L12: No need for "different".

9. P3, L13: Change "sometimes thermodynamic conditions are the main drivers, sometimes dynamic conditions" to "either thermodynamic conditions or dynamic conditions may be the main driver".

10. P8, L31: "in the Fourier space along all parameters change simultaneously" is awkward; please reword.

---

## Referee Comment (RC2) · Anonymous Referee #2 · 27 Aug 2019

General Summary: This well-written work explores the environmental and microphysical uncertainties that produce the largest variability in deep convection and hailfall characteristics. The authors make an excellent point that the impact of such uncertainties needs to be considered not only in isolation, as previous studies have largely done, but also in relationship. Hence, the work will be an excellent contribution to the literature. There are a few points about the effect of the chosen microphysical parameterization and comparison of variability to other studies that I would like clarified, but generally I support the acceptance of the article pending minor revisions.

Specific comments: 1. These results have to potential to be highly related to the choice

of microphysical scheme, and as such, this impact needs to be discussed in the paper. How much of an effect do certain choices made in the microphysical scheme have on these results – for example, could the chosen hail/graupel ice collection efficiency relationships affect the high variability found to be associated with the ice multiplication coefficient? What about the density of the rimed ice? (Frankly, I was surprised that wasn't chosen as an input parameter to vary as multiple studies have noted its importance; e.g. Morrison et al. 2015, JAS.) Most importantly, are these results transferrable to other double-moment microphysical parameterizations with a similar number of classes, or are they unique to this scheme alone?

2. The environmental condition input parameters (surface potential temperature and wind shear) are varied over a smaller range than most environmental sensitivity test studies. The authors explain this range of environmental conditions was chosen as it corresponds to typical environmental uncertainties seen in COSMO. Limiting the perturbation to that range is important, in my opinion, as it allows the work to make judgements about which model improvements are most likely to improve simulations of convection and hailfall. However, the results cannot be compared to other works examining the impact of the full range of environmental conditions that can produce hail, such as Dennis and Kumjian (2017) and Storer et al. (2010), without explicitly comparing the ranges of inputs of environmental conditions in all studies. I would like the article, especially the abstract, to emphasize that the input range of environmental conditions is only meant to encompass model uncertainty.

To that end, more information about how the variations in surface potential temperature and the scaling factor for 0-6 km shear translate to typically cited ambient environment conditions, such as CAPE and 0-6 km shear, would be helpful. The shear in particular is important given the results of Dennis and Kumjian (2017). A hodograph plot showing the range over which the shear profile was varied would be useful.

3. I'd like more information about the emulators, inputs, and training data. Pg. 8 line 6 mentions using a "choice of input combinations of the parameters" to train the

emulator. What combinators are selected and how is the choice made? How many simulations were required to train the data, and what outputs were used? How many emulations were eventually produced – one for each possible combination of input variables? Are the 10,000 realizations of vertical heating profiles produced using the same combination of input parameters and the same emulator method?

4. The discussion of the variability of the hail size distribution caused by the different input parameters focused solely on the maximum and minimum number concentrations and not the distribution of responses within those bounds. Within Fig. 6, could box and whisker plots be used to show the distribution of number concentrations within each of the three setups within a set range of size bins? That would allow the distribution of distributions, as it were, to be discussed.

To Figs. 3a and 6 I'd also like to see added the range over which N(D) and D are allowed to vary within the microphysical scheme used, for the range of ïĄő used. That would place the amount of variability in context. I'd also like to see Fig. 3b repeated with the data analyzed in Setup 3 and Fig. 6, as I feel it lets the reader more easily grasp the key ingredients in the output variability.

Minor comments: Pg 4, line 7: Do the authors feel 1 km is of a fine enough resolution for this study? From the literature, would they expect any of the results to change if this resolution were reduced?

Pg 4, line 15: A quick sentence here clarifying the difference between saturation adjustment and explicit diffusional growth would be helpful.

Pg 6, Table 2: Over what intervals were these values varied?

Pg. 7, line 11: Why is mu held constant?

Pg. 7, line 16: "chosen such that the most important parameters. . .are considered" – how were these chosen?

Pg. 9, lines 4-7: Nice description.

Pg. 9, Section 4.1, lines 15 – 5: The units of these variables need to be included. Is hail at ground and max hail at ground accumulation-based over the final 5 hours of the simulation? Are the mean and maximum values mentioned calculated in both space and time? Is precipitation rate of hail a flux of the mixing ratio through the lowest model level?

Pg. 10, Figure 1: Reorder the colors in the bar plot so they are the same order as the legend – many of them are similar shades.

Pg. 10, line 24: "to examine how the simulated storm impacts the ambient conditions" is an odd phrasing. "Ambient", to me, indicates the environmental air surrounding the convection. Diabatic heating profiles can modify this region through gravity waves and other atmospheric responses, but here the authors are focused on in-storm effects. I would reword to "examine how the heating profiles of the simulated storm change".

Pg. 11, line 7: "covering the whole parameter space" – is this the whole input parameter space?

Pg. 12, line 4: Instead of number density, should this be number concentration?

Pg. 12, line13: when referring to the "lowest number concentrations of hail", to what diameter are you referring?

Pg. 12, last two sentences: From this figure, it appears to me CCN has a larger effect than the strength of the ice multiplication.

Pg. 13, Table 3: Per Khain et al. 2011 (Atmospheric Research), the signal of CCN changes associated with hail fall switches sign around 3000 cm-3. Do the authors think their results are in line with this study?

Pg. 14, lines 3-4: I would argue the contribution due to CCN is larger in both S1 and S2 than S3.

Pg. 15, lines 34-40: See specific comment #2.

[Figure]

Pg. 16, line 10: See specific comment #2.

Pg. 15, second line 5: it seems like the authors are arguing there is a difference between "the cloud" and "the integrated amount of cloud water" in the Storer et al. (2010) study. Could they explain what his difference is?

Pg. 16, Fig. 5: I'm unable to see the different standard deviation distributions. Perhaps instead a similar plotting method as in Fig. 6, with individual lines of different styles marking the edges of the standard deviations.

Pg. 17, lines 28-29: Saturation adjustment was not one of the input parameters selected to test, so how can the authors make this claim?

Pg. 17, lines 3-4 (just before section 5.3): See specific comment #2.

Pg. 18, lines 17-18: Can this sentence be worded more clearly? Perhaps "the spread of the distributions in S2 is larger than S1, particularly for smaller diameters of hail." Can the authors comment why the uncertainty is so much larger for smaller diameters?

Pg. 19, first and second lines: The authors note the relationship between low fall velocity of graupel and high number concentrations of hail several times in the paper. Could they provide a physical explanation for this relationship?

Pg. 19, line 20: "than the inputs related to environmental conditions". . .on the scale of uncertainty seen in COSMO. See specific comment # 2.

Pg. 21, line 23-26: An excellent summary of the potential impact of this research – but it depends on the transferability of the results to other microphysical parameterizations (see specific comment #1).

Typographical: Pg. 4, line 31: Add a comma after profile. Pg. 10, Eq. 3: instead of nu/s, which denotes division, I'd use nu,s. Pg. 10 line 34: "such that"–> where

---

## Referee Comment (RC3) · Anonymous Referee #3 · 6 Sep 2019

Review of "Comparing the impact of environmental conditions and microphysics on the forecast uncertainty of deep convective clouds and hail" by Wellmann at al.

The study identifies model input parameters describing environmental conditions and cloud microphysics that lead to large uncertainties in the prediction of deep convective clouds and precipitation, by conducting statistical emulation and variance-based sensitivity analysis of the simulated deep convective clouds in an idealized setup of a cloud-resolving model. They showed some interesting results that could be useful in guiding the improvement of forecasting. However, the results could be very dependent of microphysics scheme, model setup (such as idealized vs. real, nesting vs non-nesting), and

even convective case. This discussion would be necessary. Particularly, the two moment microphysics schemes with saturation adjustment for condensation/evaporation calculation could lead to very different CCN impacts on latent heating and precipitation rate compared with more explicit microphysics schemes such as bin scheme as detailed in a review article (Fan et al., 2016, JAS). This could change the conclusion related to the diabatic heating rate. Another major problem of this manuscript is that the authors only described what the figures show, and did not interpret the results by connecting with physics properties/processes. See my specific comments for Section 4 and 5. The paper also have quite a bit confusing statements that need to be clarified. Therefore, a major revision is recommended to improve the paper before it is accepted for publication.

Specific comments

The title has a grammatic error: it should be "Comparing . . . to (or with). . ."

Abstract:

Need some detailed background about how change of environmental conditions affect deep convective cloud properties. P1, L8-9 I think the results section showed that fall speed of graupel even contributes more than the fall speed of hail. In the last sentence, suggest rewrite or add sentence to show what parameters impact hail.

Introduction:

1. P1, L16, Fan et al., JGR, 2009 and Qian et al., JGR, 2015 are the studies focusing on wind shear impacts on convective clouds. 2. P2, L5, Change the second "for" to "with". 3. P2, L22-23 The sentence "However, the impact on precipitation is not identified as the investigated clouds are non-precipitating" needs to be rewritten. I have no idea what you want to say here. 4. P2, L25, The sentence "because of its higher fall velocity immediately falls out of the cloud leading to reduced convection intensity" has grammar errors and also confusing. 5. Need to change the strong tone in some statements.,

for example, (1) "There are only a few studies including Lee et al. (2008) and Storer et al. (2010) where the effect of several parameters is analyzed", you do not need to say only a few studies since there are a significant number as far as I Know. If indeed just a few, all of them are needed to be cited here. (2) "The only previous studies of multiple interacting uncertainties in deep convective clouds are our own previous study (Wellmann et al., 2018) and Johnson et al. (2015)."

Model Setup:

1. Since the open lateral boundaries are used, need to specify how the boundaries are set up, i.e., what are used for the boundary conditions? 2. P4, L20 How did you define cloudy points? 3. P4, L34-35, the recent progress about CCN impacts on convective clouds is Fan et al., (2018, Science). 4. P5, L15-16, this is confusing, how can you specify the wind velocity to be constant in all simulations since wind is a prognostic field?

Sections 4 and 5:

1. I think some brief introduction to the case is needed before discussing the results from uncertainty quantification (UQ), which would help understand the UQ results . For example, I would like to know the relative amount of each hydrometeor mass to understand if this is a hailstorm case or not (i.e., hail mass is dominant compared with graupel mass). This would help me understand why graupel fall speed is the largest contributor to the uncertainty of integrated hydrometeor mass. 2. In both Section 4 and 5, there is a problem that the authors only describe the figures, but do not interpret the results from physics perspective. For example, in describing Fig. 1, it is better to understand why graupel fall speeds and CCN have the largest impacts on integrated hydrometeor mass but not on the hail mass? Why CCN have a large contribution to integrated hydrometeor mass but not to diabatic heating? 3. I have a hard time to physically understand the contributions shown in Figure 2. At the maximum heating around 3-6 km, the latent heating should be dominated by condensation, which

should be strong affected by CCN. Bout because saturation adjustment is used for condensation and evaporation, the CCN effect on condensation is not shown here. This problem should be discussed. In addition, How does graupel and rain fall speed contribute to the heating uncertainty? Above 10 km, the major contributors are CCN and graupel fall speed. I think it is because they affect how much amount of condensate mass are being transported to the upper levels. This kind of discussion is important to connect with cloud physics. 4. P12, L18-20 Figure 3 shows the largest contribution is graupel and hail fall speeds, which is different from what is described here. 5. P12, L20-21, need to discuss the possible physical mechanisms of how CCN affect the large hailstones. There are literature studies about this. 6. P15, the first three paragraphs, need some discussion in connecting with cloud physics to understand why. For the third paragraph, how to explain the contrasting contribution of hail fall speeds to hydrometeor mass and precipitation? 7. P17, "Dennis and Kumjian (2017) specify in their work that process rates are not an essential factor causing discrepancies in the formation of hail for different model setups", not sure what this means, since microphysical process rates directly determine the budget. 8. Section 5.3, need clearer introduction here to state the purpose of this part. I was not understanding the purpose of this part until I got to the summary (last paragraph of page 20). 9. P18, L12-21, all of the magnitudes described in these two paragraphs are different from what is shown Figure 6. For example, the maximum value plotted is 0.01 mm-1m-3, but you got values of 0.4 and 3.4 mm-1m-3 in the text. Need to check what is going on. 10. P18, L15-16, the sentence is confusing and need clarifications.

Section 6:

1. The relevant summary (the first three paragraphs) needs to be revised accordingly by adding physical explanations. 2. For "The controlling parameters of the combined input parameters are the INP concentration and the fall velocities of graupel and hail, hence a combination of parameters describing environmental conditions and micro-physical parameters", the logic of the sentence is wrong. All the parameters described

here are only microphysical parameters.

---

## Author Comment (AC1) · 12 Nov 2019

The comment was uploaded in the form of a supplement:
https://www.atmos-chem-phys-discuss.net/acp-2019-558/acp-2019-558-AC1-supplement.pdf

---

## Author Response (AR1)

We would like to thank all three reviewers for their constructive criticism of our manuscript. Their comments have helped us to improve the text.

Point-by-point replies are inserted below. The reviewers' comments are in italics, while our answers are in normal font. Where we refer to new/modified text from the manuscript, this is given in blue color.

***Anonymous Referee #1***

*This study examines the relative influence of environmental conditions and microphysical parameters on vertically integrated hydrometeor contents and precipitation values (with an emphasis on hail), in addition to their influence on heating rates. The authors use an emulator technique to reduce the number of simulations that would otherwise be required to produce the presented results. In general, the study finds that environmental conditions and microphysical properties contribute to forecast uncertainty; however, when the environment and microphysics interact with each other, the latter tends to dominate forecast uncertainty in hydrometeor contents and precipitation.*

*Overall, I have a long list of comments; most importantly, I think that the authors should provide more analysis and discussion in several of the sections (please see general and specific comments). The paper is well written, but readability could be improved by a more liberal use of commas in addition to the rewording of several sentences (please see specific comments). In addition, the authors should try to be consistent about their use of active versus passive voice. With all of this said, I think that the results are interesting and worthy of publication, and at this stage I suggest acceptance subject to major revision.*

We thank the reviewer for the constructive comments. We have addressed all comments individually below. Additionally, we have improved the language in the entire manuscript.

*Major/general comments:*

*1. Model Setup (Section 2) and Methods (Section 3): For clarity, and especially for those readers who are not familiar with the emulator technique, there should probably be more information included about the modeling approach.*

*For example, what is the total number of simulations conducted?*

We have used *15×k* input combinations to train the emulator, with *k* the number of input parameters, which is 6 in S1, 7 in S2 and 6 in S3. Furthermore, 10 simulations were added to the training datasets of S1 and S3 to increase the quality of the emulator fit. Thus, per Setup, 100 (S1 and S3) or 105 (S2) simulations were run to generate the training data. Additionally, 45 simulations with other input parameter combinations were conducted per setup for the evaluation of the emulators. In sum, the total number of simulations is 440.

This information has been added in section 3.1.

*Which "input combinations" are simulated?*

The explanation is given on p. 8: „This is ensured by the use of maximin Latin hypercube sampling (Morris and Mitchell, 1995) to select these input combinations." The combinations of input parameters used in the three setups have been added to the published dataset accompanying this study (doi:10.5445/IR/1000099232).

*How does your choice for the mean function and correlation structure influence the results?*

The choice of the linear trend for the mean function and the Matérn correlation structure have been discussed in more detail by Lee et al. (2011), and have since then be used by a number of studies (Johnson et al., 2015; Igel et al., 2018; Wellmann et al., 2018, Glassmeier et al., 2019). We have not investigated the impact of this choice in our study, and think that adding such an analysis would go beyond the scope of this manuscript. We have added these references to the text.

Glassmeier, F., Hoffmann, F., Johnson, J. S., Yamaguchi, T., Carslaw, K. S., and Feingold, G.: An emulator approach to stratocumulus susceptibility, Atmos. Chem. Phys., 19, 10191–10203, https://doi.org/10.5194/acp-19-10191-2019, 2019.

Igel, A. L., Heever, S. C., & Johnson, J. S. ( 2018). Meteorological and land surface properties impacting sea breeze extent and aerosol distribution in a dry environment. Journal of Geophysical Research: Atmospheres, 123 22– 37. https://doi.org/10.1002/2017JD027339

Lee, L. A., Carslaw, K. S., Pringle, K. J., Mann, G. W., and Spracklen, D. V.: Emulation of a complex global aerosol model to quantify sensitivity to uncertain parameters, Atmos. Chem. Phys., 11, 12253-12273, https://doi.org/10.5194/acp-11-12253-2011, 2011.

*How do you validate the emulator?*

As described in Wellmann et al. (2018), 45 additional simulations, also sampled via maximin Latin hypercube sampling, were conducted for the validation (per setup). When comparing the emulator results to the results of the validation simulations, only a small number of outliers (up to 3) outside the 95% confidence intervals are accepted. In addition, a test for robustness of the choice of the training dataset has been conducted by interchanging the training dataset with parts of the validation data. This information has been added to the text in section 3.1.

*Are the emulator results sensitive to the chosen minimum and maximum parameter values?*

Yes, the "Main effect" scales directly with the parameter range, so this selection is crucial for the interpretation of the results. This is mow stated more clearly in the Summary & Conclusions ("Note that the range of variation of these parameters is designed to mimic typical forecast errors and is therefore smaller than in earlier studies, which have encompassed a wider range of possible conditions." and "For our choices of input parameter ranges, …"). In section 2, we have added the following sentence: "Note that as the results depend crucially on the ranges over which the parameters are varied, these have to be chosen carefully and taken into account when comparing to other studies."

*What is the argument for including CCN and INP in the environmental conditions setup?*

CCN and INP concentrations are linked to aerosol concentration and type, which changes in different environments (e. g. urban compared to marine). This is different for the other microphysical parameters in S2, for which the variation spans a range of uncertainty due to e.g. different ice densities of graupel and hail, which are not clearly linked to specific conditions.

*Do the results change notably if these two microphysical properties are included only in the microphysical conditions setup?*

No, because the comparative evaluation is based on setup S3.

*2. Sensitivity Analysis for Variations of the Microphysics (S2) (Section 4): This section, which represents the bulk of the paper, generally lacks depth and therefore should contain additional insight and discussion.*

The aim of this study was not a detailed process analysis, but a general overview and a statistical quantification of the relevance of the uncertainty of various parameters. The large ensemble is not well suited for the investigation of causal relationships. Nevertheless, we have inserted more discussion on possible explanations for our results, partially based on related studies.

*For instance: [P9, L14-20] Why look at the mean for hydrometeor content, max for precipitation, and both for amount of hail?*

The output variables have to be reduced to 0 dimensions in order to be represented by the emulators. This requires averaging or selecting the maximum values. Our reasoning was that, in general, we are

interested in the variables that are linked to severe weather at the surface (as precipitation maxima and hail), but also in the in-cloud processes causing them. For this, the mean hydrometeor loads are of interest. We have added this information to the text.

*[P9, L31-32] What about snow and hail as contributors to the output uncertainties? Maybe this should say one of the largest contributors.*

We have modified the sentence as follows: "Fig. 1 reveals that of the investigated parameters, the graupel fall velocity factor $a_G$ is the largest contributor to the output uncertainties of most of the integrated hydrometeor masses".

*[P9, L39-40] Total precipitation, which is a very important quantity, seems to be affected more notably by the fall speed of graupel scaling factor than by CCN. Please comment on this.*

This is likely due to cold phase processes (riming, depositional growth) dominating precipitation formation, as was shown e.g. by Schneider et al. (2019) for cases of strong convection. Riming is more directly impacted by the graupel fall velocity than by CCN, although the latter has an indirect impact via the modification of droplet size.

At the end of section 4.1, we have removed the sentence "Contributions from the other parameters are only of minor importance" and added the following new text: „For the maximum total precipitation, the scaling factor for the fall speed of graupel, $a_G$, is also relevant. This is in line with the expectation that for for cases of strong convection, cold phase processes (including riming onto graupel) dominate precipitation formation, as was shown e.g. by Schneider et al. (2019)".

*3. Heating rates (Section 4.2): The results from this section are quite interesting; however, I feel as though it is lacking a bit in terms of analysis and discussion. It would be nice if the authors took some time to dive a little deeper. For example: [P11, L17-18] Why is the fall speed of graupel the most important at low altitudes (<2 km) and high altitudes (>10 km), which is where the graupel heating rates are very small if not zero? Is graupel present in these regions? Perhaps a figure showing vertical profiles of hydrometeor contents may help.*

This is an interesting point. We have decided not to include additional figures with the vertical profiles of hydrometeor content to save space, because they are in line with the diabatic heating rates and do not offer surprises. Below we show the hydrometeor profiles from Fig. 3 of Barrett et al. (2019), who used a near-identical model setup. We have also added a reference to this figure in the text.

[Figure]

The contribution of CCN and graupel fall speed to uncertainty of the heating rate at z>12 km can be explained as follows: In COSMO, graupel is not only produced by riming, but also by freezing of rain drops, and therefore many (actually rather small) graupel particles are present at altitudes up to 12 km. At the cloud top, the freezing occurs homogeneously and is not related to the INP concentration nor the scaling

factor introduced for INP. Therefore, the production and subsequent sublimation of ice hydrometeors at high altitudes is impacted by CCN (which impact how much rain water is produced and transported to the homogeneous freezing level), but only to a small extent by INP. The graupel fall speed impacts riming at lower levels in the cloud (again impacting how much rain water is transported to the homogenous freezing level) and in addition determines the gravitational sink of the graupel particles present at these altitudes. Snow and ice contribute to a larger extent to the latent heating by sublimation at these altitudes, but the parameters determining their fall speeds have not been considered here.

We have added the following more concise explanation to the text: "Above [10 km], the output uncertainty of the total heating rate is dominated by the CCN concentration and the fall velocity of graupel. This is probably linked to the indirect effect of CCN and riming efficiency on the amount of supercooled water transported to the homogeneous freezing level. Furthermore, graupel is produced at these levels in our model as a result of the freezing of rain drops, and the graupel fall speed factor thus impacts the gravitational sink of the (small) graupel particles present at these altitudes."

For the explanation of the graupel impact on heating rates at z < 2 km, we refer to the analysis of vertical profiles of hydrometeors and process rates in Barrett et al. (2019), who used a very similar configuration and base case setup of the COSMO model. At lower levels, rain evaporation (together with cloud water condensation) is the main term in the latent heating rain. As shown by Barrett et al. (2019), roughly 50% of the surface rain originates from cold rain processes involving riming. Therefore here the graupel (and also hail) fall speed parameters contribute substantially to the uncertainty of the latent heating rate at levels below 2 km, although there is no graupel present at these altitudes.

To include this into the text, we have added/reformulated the following sentences: "In the height between 3 km and 4 km there are also major contributions from the fall velocity of rain $a_R$. Below, coinciding with the largest cooling due to the evaporation of rain, $a_G$ is again the major driver of uncertainty. As shown by Barrett et al. (2019), roughly half of the surface rain in this model setup originates from cold rain processes involving riming. Therefore here the graupel (and also hail) fall speed parameters contribute substantially to the uncertainty of the latent heating rate at levels below 2 km, although there is no graupel present at these altitudes."

*[P11, L18-19] Where the fall speed of rain plays a notable role in the main effect (between 3 and 4 km), the magnitude of the rain heating rate is only a small fraction of the magnitude of the total heating rate. Does this suggest that, in general, the model physics is more uncertain about rain evaporation processes than cloud condensation processes?*

This is possibly linked to the saturation adjustment used for cloud condensation, which is thus insensitive to droplet number and size, while rain evaporation is treated as a time- and size-dependent process. We have added the following sentences: "CCN contributes only modestly to uncertainty at these levels, although the heating rate by condensation is very strong here. This is probably linked to the fact that a saturation adjustment scheme is used for cloud condensation, which is thus insensitive to droplet number and size."

*[P12, L2-3] Can you speculate as to why the CCN concentration and fall speed of graupel dominate the total heating rate output uncertainty at high altitudes?*

As explained above, we think that this is an indirect effect. It is important to note that the parameters are not changed level by level, but for the entire setup; thus, changes in the graupel fall speed affect how much riming occurs in the lower levels of the cloud and thereby controls how much liquid water is transported to higher altitudes (and causes latent heating there by freezing or evaporation). Additionally, the graupel fall speed controls the gravitational sink of graupel at the cloud top. Similarly for the CCN concentration, which (among other things) impacts the efficiency of warm rain formation at lower levels and therefore the rain water content at the homogenous freezing level.

*4. Hydrometeor masses and precipitation (Section 5.1): In general, this section would benefit from a deeper (and more quantitative) analysis. Figure 4 is really nice for visual comparison; however, can the numerical values be put into a table (perhaps in an appendix or a supplement) for a more quantitative comparison?*

Yes, the numerical values have been added as Tables A1 and A2 in the appendix.

*Also, to minimize reader confusion, I recommend not putting a circle in areas where the input parameter was not part of the emulator simulation. For instance, under S2 for shear, under S1 for graupel fall speed, etc.*

In a revised version of the figure, the circles for these input parameters have been removed.

*Some comments and questions about the section text: [P14, L4-6] When referring to the trend for precipitation output, what about for total precipitation rate, which appears to be different? Is this important? Again, a table would help clarify these comparisons.*

The reviewer is correct that the CCN contribution to the main effect is similar in S1 and S2, but smaller in S3 for the maximum total precipitation rate, while it decreases steadily from S1 to S2 to S3 for the maximum accumulated total precipitation. However, we don't think this effect should be overemphasized, because it could be due to the choice of showing the maximum and not the mean, such that there is some noise in the results.

*[P15, L3] When referring to the main effect of the INP concentration, what is the physical interpretation? The influence of uncertainty in the INP concentration is muddled when the uncertainty in the other (individual? some? all?) microphysical parameters are introduced? Please elaborate.*

We hypothesize that secondary ice formation can drown the effect of primary ice formation when it's very efficient. Similarly, when the graupel fall speed is large, this can result in very efficient riming and therefore consumption of supercooled liquid, again reducing the importance of heterogeneous ice nucleation. We have added these thoughts to the text:

"Thus, the main effect of the INP concentration is smaller if other microphysical parameters are used as input, possibly because other ice phase processes (secondary ice formation, riming) can suppress the sensitivity of a cloud to primary ice formation."

*Minor/specific comments:*

*1. P1, L15-17: Maybe reword to make more clear that you are emphasizing environmental parameters and microphysical parameters. Also, please separate the citations to better associate with these two different aspects of forecasting convective clouds.*

Rephrased to: "Thus, numerous studies have been published on simulating deep convective clouds. These have investigated how environmental parameters like wind shear (e. g. Weisman and Klemp, 1984, Lee et al., 2008), and the aerosol environment, which determines the CCN concentration (e. g. Lee et al, 2008; Rosenfeld et al., 2008, Fan et al., 2013), affect the clouds in these simulations."

*2. P2, L3: What are the different choices of the trigger?*

We have added this to the text: "a warm bubble, a cold pool or a bell-shaped mountain ridge".

*3. P2, L5-6: Please add references for the Morrison and Thompson schemes.*

Done.

*4. P2, L6-7: Which aspects of the parameterizations are most influential?*

We have inserted this information by extending this sentence as follows: "They find that the use of the two schemes causes larger differences than the changes in the number concentration, primarily because of the representation of autoconversion of cloud water to rain and of cloud ice to snow."

*5. P2, L8: Individual parameters such as?*

This sentence was confusing and has been removed.

*6. P4, L7: Horizontal resolution or grid spacing?*

We meant grid spacing. This has been rephrased.

*7. P4, L9: Can you provide approximate vertical grid spacings in the layer(s) of interest?*

We have added this information to the text:

"These levels follow the transformation given in Gal-Chen and Somerville (1975) such that they are denser near the ground and further apart with increasing height (approximately 300 m vertical distance at 5 km altitude and 400 m vertical distance at 10 km altitude). Variables are written out and analysed on interpolated z-levels with 250m vertical distance up to 3km and 500m vertical distance above."

*8. P4, L9: Do the open boundary conditions cause any mass conservation issues?*

The mass of the individual tracers in the domain is not conserved, because the considered air mass eventually leaves the domain. However, we have selected the domain size such that the cloud does not reach the domain boundaries within the considered simulation time.

We have added the following sentence to the text: "During this simulation period, the clouds do not reach the boundaries of the domain."

*9. P4, L20: How are cloudy grid boxes defined?*

We have specified this more clearly now:

"We consider only cloudy grid points (where the vertically integrated content of any hydrometeor type is >0) in our analysis of the vertically integrated hydrometeor contents."

*10. P4, L26: Which "two former studies"?*

We have replaced this part of the sentence by "… using only the key inputs of the setup with variation of environmental conditions and the new setup with variations in microphysical parameters, in order to enable a comparison of the relative importance of environmental and microphysical uncertainties for model output uncertainty."

*11. P5, Table 1: The fourth input says "potential temperature at the ground", but the text says "vertical temperature profile". Please clarify.*

The Weisman & Klemp (1982) temperature profile follows the function

$\theta(z) = \theta_0 + (\theta_{tr} - \theta_0)(\frac{z}{z_{tr}})^{\frac{5}{4}}$   for $z$ below the tropopause height $z_{tr}$, which has a fixed potential temperature $\theta_{tr}$.

Thus, $\theta_0$ is the potential temperature at the ground, but it impacts the entire tropospheric profile. We have modified the text by inserting "This variation of $\theta_0$ impacts the entire tropospheric profile and corresponds to a change of CAPE …".

Furthermore, we have inserted Weisman & Klemp's equation for the temperature profile.

*12. P5, L6: Perhaps "maritime" should be changed to "clean" because marine cloud can be polluted.*

Done.

*13. P5, L17-18: This sentence is confusing…does it turn toward the west until a straight easterly flow is reached or does it turn toward the east until a straight westerly flow is reached?*

We have added the equation for the profile of the wind direction to remove any ambiguities.

*14. P5, L20: Please state explicitly the wind direction bounds at the surface.*

As requested, we have added this information to the text:

"Here, we vary $F_{shear}$ only between 0.3333 and 0.6666, corresponding to a wind direction at the ground between 210° and 240°, which reflects the typical error range of the operational COSMO forecast of the wind direction (Felix Fundel, personal communication, 2017)."

*15. P5, L20: Please add a citation for this statement.*

Unfortunately, there is no citable peer-reviewed document for this statement, but we have specified our source as "(Felix Fundel, personal communication, 2017)".

*16. P5, L23: Please add a citation for this statement.*

Again, there is unfortunately no citable peer-reviewed document for this statement, but we have specified our source as "(Felix Fundel, personal communication, 2017)".

*17. Table 2: Please add the symbol/abbreviation for the various parameter inputs (where necessary).*

We have added the variable names $a_R$, $a_G$ and $a_H$ in Table 2.

*18. P7, L11: Why not also vary μ?*

We follow here Igel and van den Heever (2017a,b). With the dispersion parameter μ=1/3, the general gamma distribution of Seifert and Beheng (2006a) for the cloud droplet mass reduces to a (simple) gamma distribution for the drop diameter, which is the type of distribution used by Igel and van den Heever (2017a,b). We have not been able to find any studies on the spread of the dispersion parameter in observations. An earlier study with the Seifert and Beheng (2006a) cloud scheme (Noppel et al., 2010) has also changed μ, but we think that this additional parameter would not add another dimension of uncertainty to our ensemble.

We have added a reference for Igel and van den Heever (2017a).

Igel, A.L. and S.C. van den Heever, 2017: The Importance of the Shape of Cloud Droplet Size Distributions in Shallow Cumulus Clouds. Part I: Bin Microphysics Simulations. J. Atmos. Sci., 74, 249–258, https://doi.org/10.1175/JAS-D-15-0382.1

*19. P7, L32: Please change "data" to "output".*

"output" wouldn't fit well here because at this point the model output is the input to the variance-based sensitivity analysis. Therefore we'd like to remain with the neutral term "data".

*20. P8, L20-21: Can you comment on the errors that are associated with this prediction?*

In response to one of the major comments, we have added information on the validation of the emulator. This also answers the question about the possible error of the prediction:

"Once an emulator is constructed, it needs to be validated to ensure an accurate estimation of the model output (Basots and O'Hagan, 2009). For this, an additional 45 simulations with other input parameter combinations were conducted per setup. When comparing the emulator results to the results of the validation simulations, only a small number of outliers (up to 3) outside the 95% confidence intervals are accepted. In addition, a test for robustness of the choice of the training dataset has been conducted by interchanging the training dataset with parts of the validation data. The validated emulator is then able to predict (with a certain error as constrained by the validation) the output at all points in the multidimensional parameter uncertainty space that were not included in the training set and thus replaces the costly simulations of the NWP model."

*21. P9, L16: Contents or vertically integrated contents?*

This sentence has been changed to "The spatial and temporal mean is taken for the considered vertically integrated hydrometeor contents."

*22. Figure 2: Is it possible to coordinate the y-axes of the two panels so that a direct comparison in the vertical is easier?*

This proved to be technically difficult because the main effect is diagnosed on more densely spaced levels below 3 km (250 m) than above (500 m). The bars would become unreadable in the lower levels if this was taken into account. Therefore we have kept the plot as it was, but have added a note of caution to the caption: "Note the different axis tick spacing below and above 3 km."

*23. P11, L13-15: While this is true, perhaps note that the total heating rate does not decrease as rapidly as the rate due to cloud water because graupel and ice heating rates are at a maximum between about 8 and 10km.*

The reviewer is correct. We have added more emphasis on the positive contributions from graupel and ice by changing the next sentence to "At higher altitudes, there are additional positive contributions from the formation of graupel and ice."

*24. P12, L31-33: The impact of the fall speed of hail and the strength of ice multiplication is mentioned as being important, but what about the role of the fall speed of graupel? This seems quite important for small hail. Moreover, can you speculate as to why CCN concentration becomes more important at larger hail diameters?*

In COSMO, hail can form through two different processes: riming of graupel and freezing of rain. This may explain the different sensitivities to CCN and $a_G$ of different parts of the size distribution. However, a more detailed process analysis is not possible within our framework. We have added the following text:

"At the largest considered diameters, an increased contributions from the CCN concentration comes into play, while smaller diameters are significantly impacted by the graupel fall speed. This may be linked to the two formation pathways of hail in COSMO, namely through freezing of rain (of which the size is impacted by the CCN concentration) and through riming of graupel."

*25. Table 3: What is the difference between assigning the input parameters "lower" and "higher" values as opposed to "min" and "max" values (as in Tables 1 and 2)? Also, why are the values used in this experiment different from those used for S1 and S2? Is the input listed as "potential temperature" at the surface?*

There was an error in the parameter values for $F_{shear}$ in Table 3. The correct values are 0.3333 and 0.6666, thus the same as in Table 1. This has been corrected.

As discussed in Wellmann et al. (2018) and in section 2.1, the parameter range for $F_{shear}$ and $\theta_0$ has been restricted to the amplitude of typical forecast errors. For the other parameters, we have tried to encompass all possible values when defining the model setups, but have restricted them to a narrower range of more likely values based on literature for the analysis of the hail size distribution. This choice is certainly, to some extent, subjective.

We have added the following text: "Hereby, the outer bounds of the environmental parameters $F_{shear}$ and $\theta_0$ from S1 are taken as "-" and "+", as they are already limited to the typical range of forecast errors. For all other parameters, the lower and higher values are subjectively chosen to be representative, but not extreme, and encompass therefore a smaller range than examined in S1, S2 and S3."

*26. P14, L1 (below Fig. 4): Can you parenthetically reference the CCN and INP concentrations here to help the reader?*

Done.

*27. P15, L4-7: Maybe note that, in general, uncertainties in wind shear (which is likely not uncommon in numerical weather prediction) do not have a notable impact on the output variables examined here with the exception of integrated rain water content (rain water path).*

As suggested, we have added this information and reformulated the relevant sentences:

"The behavior of the wind shear is quite consistent for the considered output variables. Its contribution is in general small, except if the intergrated rain water content is the target output variable. It is always larger in S1 than in S3, meaning that the wind shear has a larger impact on the output uncertainty, if only the environmental conditions are varied."

*28. P15, L6: Perhaps note that the impact of theta is already relatively low.*

As suggested, we have changed this sentence to: "Similarly, the (already small) impact of $\theta_0$ …"

*29. P15, L31-32 ("The remaining input parameters […]"): I am not sure that I understand this sentence. Please clarify.*

This sentence has been changed to: "The other input parameters ($\Delta T$, radius, $a_R$, the ice multiplication factor and the shape parameter) are only used in one of the setups so that a direct comparison of different setups is not possible."

*30. P15, L34-35 ("in particular from the fall velocity of graupel for the hydrometeor masses and from the fall velocity of hail for precipitation"): This seems like an important finding; please italicize for emphasis.*

Rather than italicizing this statement, we now emphasize our findings further by iterating it in the abstract and the conclusions:

"The microphysical parameters, especially the fall velocities of graupel and hail, lead to larger uncertainties in the output of integrated hydrometeor masses and precipitation variables."

"The study combining both sets of input parameters shows a large contribution by the fall velocity of graupel to the output uncertainty of the hydrometeor loads, and by the fall velocity of hail to the output uncertainty of the precipitation variables."

*31. P15, L35-36: So, are you able to say that uncertainties in the selected environmental conditions are muddled by uncertainties in the selected microphysical parameters?*

Our point is here that it depends what one is looking at. For the diabatic heating rates, the uncertainties in environmental conditions dominate, but for the hydrometeor and precipitation variables, the uncertainties in microphysical variables prevail.

*32. P15, L50-51 ("In their study […]"): This sentence is a bit confusing; please reword and/or flesh out.*

This paragraph has been reworded as follows:

The impact of CAPE on deep convection is analyzed by Storer et al. (2010). In their study, the updraft strength and the total accumulated precipitation are very sensitive to changes in CAPE, while the integrated amount of cloud water does not depend strongly on CAPE. Furthermore, they conclude that the impacts of CAPE and CCN concentration can be comparable.

*33. P16, L7: To which aerosol effect(s) are you referring?*

"aerosol effect" has been replaced by "impact of CCN variation".

*34. P16, L8-11: Can you be more specific about the Fan et al. (2013) results? Changes up to 25%? 25% on average? Is their range of CCN concentration similar to the one used here? Also, for the Yang et al. (2017) paper, what range of CCN concentration was tested? This is important when comparing previous results to results from the current study.*

This paragraph has been revised to include more information on the parameter ranges and results of the cited studies.

"With respect to the impact of CCN variation, our findings are in qualitative agreement with the works of Fan et al. (2013) and Yang et al. (2017), for instance. Fan et al. (2013) find an increase of approximately 30% of the upper tropospheric cloud cover due to changes of the CCN concentration from 280 to 1680 cm$^{-3}$ (which is smaller than our parameter range). Yang et al. (2017) find clear differences in the vertically integrated condensate mixing ratio, such as an increase of ice from 6 to 18 g kg$^{-1}$, for increasing CCN from 300 to 5000 cm$^{-3}$ (similar to our parameter range). This is comparable to the significant influence of the CCN concentration on the output uncertainty of the hydrometeor contents found here."

*35. Figure 5: For the left panel, please mark the 0 K/h value to delineate between negative and positive values. Also, for both panels, is it possible to make the lines thicker in this figure (as in Fig. 2)?*

We have marked the 0 K/h line and have made the lines thicker in the left panel. In the right panel, there is too much overlap among the different lines, such that the readability would be reduced if the lines were thicker.

*36. P17, L17-18: When talking about the saturation adjustment in the microphysics scheme, how are you able to make this conclusion? Please elaborate.*

We didn't mean that the saturation adjustment would cause the uncertainty, but wanted to iterate that the condensation is parameterized via a saturation adjustment scheme. To clarify this, these two sentences have been reformulated:

"Condensation of cloud water, which is a substantial contributor to the total heating rate in the lower and middle troposphere, is parameterized via a saturation adjustment scheme in our model. Nevertheless, it yields a large contribution to output uncertainty of the diabatic heating in all three setups. This effect might be even larger if a time-dependent treatment of condensation was used."

*37. P17, L33-34: When talking about the emulator-predicted size distributions, are you able to comment on the expected uncertainty in your results?*

As discussed above (major comment #1 and minor comment #20), during the validation of the emulators it was required to meet the simulated values within the 95% confidence intervals.

*38. Figure 6: As in Fig. 5, are you able to make the lines thicker?*

The figure has been modified as suggested.

*39. Size distribution of surface hail (Section 5.3): For the analysis in this section, please use line color and style to help clarify to which distribution you are referring.*

As suggested, we have added "(continuous blue line)", "(dashed blue line)" etc. to the text.

*40. P18, L5: When referring to the value of 0.4 mm-1 m-3, this number does not seem to correspond with the y-axis. Am I missing something here?*

Thanks for pointing this out. There was a mistake in the text, while the figure was correct. This has been corrected.

*41. P18, L7: Similar to comment #40, when referring to the value of 3.4 mm-1 m-3, this number does not seem to correspond with the y-axis.*

See above.

*42. P18, L7-9: This sentence is confusing. Perhaps say something like: "when relatively high (low) values of theta and high (low) concentrations of CCN and INP are present, low (high) hail number concentrations result".*

We wanted to stay closer to our original wording, but have modified this sentence as follows: "For this setup (in which the environmental conditions are modified), the controlling parameters are the CCN and INP concentrations and $\theta_0$. Low number concentrations of hail arise for higher values of these parameters and high number concentrations of hail for lower values."

*43. P18, L10-12: This could probably be stated more simply in one sentence. For example, something like: "For S2, the maximum and minimum of the hail size distribution is larger and smaller than that for S1, leading to a larger spread in the distributions."*

We have modified the sentence as follows:

"For S2, the low (dashed red line) and high (continuous red line) hail size distributions are smaller and larger, respectively, than those for S1, leading to a larger spread in the distributions."

*44. P19, L3-10: What about for larger diameters? It is especially interesting that the number concentrations for the minimum size distributions are largest for S3. This suggests that, individually and for the lower bound, S1 and S2 do not produce large number concentrations, but if the environmental and microphysical conditions are combined (S3), then there is an enhancement. Please comment on this notable difference.*

There are two factors which make us cautious not to overinterpret this result: (a) the hail size distribution was not a target parameter when selecting the subset of parameters from S1 and S2 to be included in S3. Thus, maybe the size distribution is more sensitive to parameters in S1 and S2 which were not considered in S3. (b) the shaded area, delimited by the dashed and continuous lines, is only based on 64 combinations of parameter values. Possibly one could find more extreme hail concentrations for other parameter combinations.

For these reasons, we prefer not to speculate on this feature of the plot in the manuscript.

*45. P19, L8-10: So, can you speculate as to what this means physically?*

This sentence was a mere semantic explanation. We have omitted it now and have instead specified:

"Moreover, the controlling parameters identified in S3 include parameters from both environmental conditions (INP) and microphysics ($a_G$, $a_H$)."

*46. P19, L12-13 ("While the microphysical input parameters mainly determine the maximum number concentration, the environmental conditions substantially influence the minimum number concentration"): This seems like an important finding; please italicize for emphasis.*

We think that this statement, which is also repeated in a similar form in the conclusions, is already emphasized enough.

*47. P20, L17-27: There is a large body of literature, some of which is referenced in the introduction, that focuses on the so-called "aerosol invigoration" hypothesis. Can you relate the work presented here to previous work?*

Note that while we find that CCN is a very important parameter for both hydrometeor load and precipitation, this does not yet give a sign of the dependence. Indeed, we have been able to find configurations in which CCN increases cloud water content as well as others in which it decreases cloud

water content. A further analysis with different target variables (e.g. cloud water content in different evolution stages of the cloud) would be interesting. However, we think that this discussion is out of the scope of this manuscript.

*48. P21, L19-20: What about also a revised parameterization of the fall speed of graupel?*

We have incorporated this suggestion and have changed the sentence to "… in particular a revised parameterization of the fall velocity of graupel and hail".

*Grammatical/wording recommendations:*

*1. The authors interchange hydrometeor "content" and "mass" throughout the text. Which parameter is actually shown? Please be consistent.*

To be precise, the "hydrometeor mass content" (in kg m$^{-3}$) was meant. We have changed the wording throughout the manuscript to either "mass content", or simply "content" where the is no ambiguity.

*2. P2, L4-5: Awkward sentence; please reword.*

Sentence changed to "In addition to thermodynamic profiles and environmental conditions determining the formation and structure of deep convective clouds, also microphysical parameterizations have been shown to play a role."

*3. P2, L5: Change "three cloud types for" to "three cloud types using".*

Done.

*4. P2, L19: Maybe "Additional relevant" instead of "Further relevant"?*

Done.

*5. P2, L28: Change "Moreover, field studies indicate that fall speeds of hydrometeors are observed in a broad range of velocities." to "Moreover, field study observations indicate that hydrometeors may have a broad range of fall velocities."*

Done.

*6. P2, L30-32: Awkward sentence; please reword.*

Changed to "Gilmore et al. (2004) and Posselt and Vukicevic (2010) vary both the fall speeds and the densities of hail/graupel and snow, and find that these parameters impact the amount of precipitation significantly."

*7. P3, L10-12: Awkward sentence; please reword.*

Done.

*8. P3, L12: No need for "different".*

We have changed as suggested to "wide range of ambient conditions".

*9. P3, L13: Change "sometimes thermodynamic conditions are the main drivers, sometimes dynamic conditions" to "either thermodynamic conditions or dynamic conditions may be the main driver".*

Sentence changed to "In reality, severe convective storms form in a wide range of ambient conditions, where either thermodynamic conditions or dynamic conditions may be the main driver, leading to different organizational forms of the storms."

*10. P8, L31: "in the Fourier space along all parameters change simultaneously" is awkward; please reword.*

We have removed "along [which] all parameters change simultaneously."

*Anonymous Referee #2*

*General Summary: This well-written work explores the environmental and microphysical uncertainties that produce the largest variability in deep convection and hailfall characteristics. The authors make an excellent point that the impact of such uncertainties needs to be considered not only in isolation, as previous studies have largely done, but also in relationship. Hence, the work will be an excellent contribution to the literature. There are a few points about the effect of the chosen microphysical parameterization and comparison of variability to other studies that I would like clarified, but generally I support the acceptance of the article pending minor revisions.*

*Specific comments:*

*1. These results have to potential to be highly related to the choice of microphysical scheme, and as such, this impact needs to be discussed in the paper. How much of an effect do certain choices made in the microphysical scheme have on these results – for example, could the chosen hail/graupel ice collection efficiency relationships affect the high variability found to be associated with the ice multiplication coefficient? What about the density of the rimed ice? (Frankly, I was surprised that wasn't chosen as an input parameter to vary as multiple studies have noted its importance; e.g. Morrison et al. 2015, JAS.) Most importantly, are these results transferrable to other double-moment microphysical parameterizations with a similar number of classes, or are they unique to this scheme alone?*

We have not repeated our study with a different microphysics scheme, but agree with the reviewer that a certain dependence on the parameterizations is to be expected. However, our main aim was not to emphasize the impact of a specific parameter, but rather to weigh the relevance of environmental versus microphysical uncertainty. We expect that this result is less dependent on the microphysics scheme.

We have added the following paragraph to the conclusions: "It can be expected that our results (in particular regarding the microphysical parameters) depend to some extent on the microphysics scheme of our model. However, the overarching aim of this study was not to emphasize the impact of a specific parameter, but to quantify the relevance of environmental versus microphysical uncertainty in general. We expect that these results are less dependent on the microphysics scheme."

Considering the rimed ice density as a possible parameter, we expect that this effect would be closely linked to variation in the graupel and hail fall speed, which we have varied in our study.

*2. The environmental condition input parameters (surface potential temperature and wind shear) are varied over a smaller range than most environmental sensitivity test studies. The authors explain this range of environmental conditions was chosen as it corresponds to typical environmental uncertainties seen in COSMO. Limiting the perturbation to that range is important, in my opinion, as it allows the work to make judgements about which model improvements are most likely to improve simulations of convection and hailfall. However, the results cannot be compared to other works examining the impact of the full range of environmental conditions that can produce hail, such as Dennis and Kumjian (2017) and Storer et al. (2010), without explicitly comparing the ranges of inputs of environmental conditions in all studies. I would like the article, especially the abstract, to emphasize that the input range of environmental conditions is only meant to encompass model uncertainty.*

We agree with the reviewer that this is an important point and that our results cannot be directly compared to the mentioned studies. We have rephrased the last sentence of the abstract as follows: "In contrast, variations in the environmental parameters – the range of which is limited to represent model uncertainty – mainly affect the vertical profiles of the diabatic heating rates."

Additionally, we have added a sentence in the Summary & Conclusions section: "First, a set describing environmental conditions such as potential temperature and vertical wind shear was used. Note that the range of variation of these parameters is designed to mimic typical forecast errors and is therefore smaller than in earlier studies, which have encompassed a wider range of possible conditions."

*To that end, more information about how the variations in surface potential temperature and the scaling factor for 0-6 km shear translate to typically cited ambient environment conditions, such as CAPE and 0-6 km*

*shear, would be helpful. The shear in particular is important given the results of Dennis and Kumjian (2017). A hodograph plot showing the range over which the shear profile was varied would be useful.*

We have now inserted the equations for the profile of the wind direction and the potential temperature. In addition, we give the bounding values for CAPE and for the wind direction at the ground. (The wind direction at 6 km is 270° in all simulations.) We have decided not to include plots of the hodographs in order to limit the number of figures.

*3. I'd like more information about the emulators, inputs, and training data. Pg. 8 line 6 mentions using a "choice of input combinations of the parameters" to train the emulator. What combinators are selected and how is the choice made? How many simulations were required to train the data, and what outputs were used? How many emulations were eventually produced – one for each possible combination of input variables? Are the 10,000 realizations of vertical heating profiles produced using the same combination of input parameters and the same emulator method?*

Following Johnson et al. (2015) and Wellmann et al. (2018), we have used $15 \times k$ input combinations to train the emulator, with $k$ the number of input parameters, which is 6 in S1, 7 in S2 and 6 in S3. Furthermore, 10 simulations were added to the training datasets of S1 and S3 to increase the quality of the emulator fit. Thus, per Setup, 100 (S1 and S3) or 105 (S2) simulations were run to generate the training data. Additionally, 45 simulations with other input parameter combinations were conducted per setup for the evaluation of the emulators. This information has been added in section 3.1. The output variables are the ones shown in Figs. 1, 2 and 3: 6 hydrometeor loads, 5 precipitation variables, total diabatic heating rate on 35 vertical levels, and hail size distribution in 10 size bins (i.e. 56 emulations per setup, in total 118 emulations). Regarding the selection of combinations of input parameters, see p. 8: „This is ensured by the use of maximin Latin hypercube sampling (Morris and Mitchell, 1995) to select these input combinations." The combinations of input parameters used in the three setups have been added to the published dataset accompanying this study.

The 10,000 realizations have been produced with the emulators that were derived the same way as the emulators for the hydrometeor loads and precipitation. For clarification, the first paragraph of section 4 has been reformulated: "In the analysis, we consider several output variables for which emulators are derived as described above. These output variables, including vertically integrated hydrometeor contents, precipitation, diabatic heating rates and the size distribution of surface hail, will be described in more detail in this section."

*4. The discussion of the variability of the hail size distribution caused by the different input parameters focused solely on the maximum and minimum number concentrations and not the distribution of responses within those bounds. Within Fig. 6, could box and whisker plots be used to show the distribution of number concentrations within each of the three setups within a set range of size bins? That would allow the distribution of distributions, as it were, to be discussed.*

For Fig. 6, the spread is based on 64 parameter combinations with characteristic high and low values, not on 10,000 evenly distributed combinations. The reason for this difference to Fig. 5 is that our aim was to attribute the minimum and maximum hail size distributions to specific parameter distributions. Therefore box and whisker plots would not be well suited here.

*To Figs. 3a and 6 I'd also like to see added the range over which N(D) and D are allowed to vary within the microphysical scheme used, for the range of ïA؛o˝ used. That would place the amount of variability in context.*

Note that the shape parameter was varied for the droplet size distribution, not for the hail size distribution. Two size distributions with equal mass and shape parameters of $\nu=0$ and $\nu=8$ are shown in the below figure, taken from Wellmann (2019). We have decided not to include this figure, because the analytical equation is included and the plot can be easily produced.

[Figure]

*I'd also like to see Fig. 3b repeated with the data analyzed in Setup 3 and Fig. 6, as I feel it lets the reader more easily grasp the key ingredients in the output variability.*

[Figure]

The figure requested by the reviewer is published in Wellmann (2019), see above. We have decided not to include this figure (and the equivalents of Fig. 1, 2a, 2b and 3a) for setup S3 in order to keep the manuscript concise. Instead, we focus our analysis of S3 entirely on the relative importance of environmental and microphysical parameters and the comparison to S1 and S2, which required different plots (Figs. 4, 5 and 6).

*Minor comments: Pg 4, line 7: Do the authors feel 1 km is of a fine enough resolution for this study? From the literature, would they expect any of the results to change if this resolution were reduced?*

Many realization of the WK idealized supercell case have been published using a 1 km grid (e.g., Seifert & Beheng, 2006b). Huang et al. (2018) showed in simulations of this case with WRF that precipitation and hydrometeor content are very similar with 1 km grid spacing as with 200 m grid spacing. Potvin and Flora (2015) concluded that 1 km grid spacing produces useful forecasts despite of remaining errors (e.g. in the timing of the storm evolution). As internal cloud dynamics were not the aim of this study, we believe that this resolution is sufficient for our purpose. We have added the following sentence:

"This grid spacing was shown to be sufficient for the simulation of precipitation and hydrometeor content of idealized supercells, although vertical transport and timing differ from simulation at higher resolutions (Potvin and Flora, 2015; Huang et al., 2018)."

Huang, W., J. Bao, X. Zhang, and B. Chen, 2018: Comparison of the Vertical Distributions of Cloud Properties from Idealized Extratropical Deep Convection Simulations Using Various Horizontal Resolutions. Mon. Wea. Rev., 146, 833–851, https://doi.org/10.1175/MWR-D-17-0162.1

Potvin, C.K. and M.L. Flora, 2015: Sensitivity of Idealized Supercell Simulations to Horizontal Grid Spacing: Implications for Warn-on-Forecast. Mon. Wea. Rev., 143, 2998–3024, https://doi.org/10.1175/MWR-D-14-00416.1

Seifert, A. & Beheng, K., 2006: A two-moment cloud microphysics parameterization for mixed-phase clouds. Part 2: Maritime vs. continental deep convective storms. Meteorol. Atmos. Phys. 92: 67. https://doi.org/10.1007/s00703-005-0113-3

*Pg 4, line 15: A quick sentence here clarifying the difference between saturation adjustment and explicit diffusional growth would be helpful.*

We have rephrased the sentence as follows: „Furthermore, the model uses the two-moment bulk microphysics scheme by Seifert and Beheng (2006a), including a saturation adjustment approach (i.e. bringing relative humidity back to exactly 100% within one time step when supersaturation with respect to water occurs), predicting both the mass mixing ratios and the number densities of six hydrometeor classes (cloud droplets, rain, cloud ice, snow, graupel and hail)."

*Pg 6, Table 2: Over what intervals were these values varied?*

There are no fixed intervals. For the emulator construction, an algorithm implementing maximin Latin hypercube sampling (Morris and Mitchell, 1995) was used to select these input combinations. The combinations of input parameters used in the three setups have been added to the published dataset accompanying this study (doi:10.5445/IR/1000099232).

*Pg. 7, line 11: Why is mu held constant?*

We follow here Igel and van den Heever (2017a,b). With the dispersion parameter $\mu=1/3$, the general gamma distribution of Seifert and Beheng (2006a) for the cloud droplet mass reduces to a (simple) gamma distribution for the drop diameter, which is the type of distribution used by Igel and van den Heever (2017a,b). We have not been able to find any studies on the spread of the dispersion parameter in observations. An earlier study with the Seifert and Beheng (2006a) cloud scheme (Noppel et al., 2010) has also changed $\mu$, but we think that this additional parameter would not add another dimension of uncertainty to our ensemble.

*Pg. 7, line 16: "chosen such that the most important parameters...are considered" – how were these chosen?*

For clarification, we have added/modified this sentence:

"Based on the results of the sensitivity analysis for hydrometeor and precipitation variables in setups S1 and S2, where the sets of environmental conditions and the cloud microphysics parameters are treated separately (Fig. 5 of Wellmann et al. (2018) and Fig. 1 of this manuscript), the input parameters of this combined Setup 3 (S3) are chosen such that the most important parameters of both environmental conditions and microphysics (those that contribute most to output uncertainty across the selected output variables) are considered in addition to the CCN and INP concentrations."

*Pg. 9, lines 4-7: Nice description.*

Thank you.

*Pg. 9, Section 4.1, lines 15 – 5: The units of these variables need to be included. Is hail at ground and max hail at ground accumulation-based over the final 5 hours of the simulation? Are the mean and maximum values mentioned calculated in both space and time?*

The units are kg m$^{-2}$ for all vertically integrated hydrometeor contents, kg m$^{-2}$ for the maximum accumulated total precipitation and kg m$^{-2}$ s$^{-1}$ for the mean and maximum hail at ground per 15 minutes as well as for the maximum precipitation rate of hail and the maximum total precipitation rate.

Maximum and mean values are taken both in space and time (except for accumulated total precipitation, which is integrated over time).

We have added this information to the text, such that it now reads

"The spatial and temporal mean is taken for the considered vertically integrated hydrometeor contents (all in kg m$^{-2}$). The set of considered precipitation variables include the amount of hail at the ground per output interval of 15 minutes, the precipitation rate of hail and the total precipitation rate (all in kg m$^{-2}$s$^{-1}$) and the accumulated total precipitation (in kg m$^{-2}$). Precipitation is analyzed similarly to the hydrometeor

masses, but maximum values in space and time are considered instead of mean values. An exception is the amount of hail at the ground, for which both mean and maximum values are analyzed."

*Is precipitation rate of hail a flux of the mixing ratio through the lowest model level?*

No, this is the hail reaching the surface.

*Pg. 10, Figure 1: Reorder the colors in the bar plot so they are the same order as the legend – many of them are similar shades.*

Done.

*Pg. 10, line 24: "to examine how the simulated storm impacts the ambient conditions" is an odd phrasing. "Ambient", to me, indicates the environmental air surrounding the convection. Diabatic heating profiles can modify this region through gravity waves and other atmospheric responses, but here the authors are focused on in-storm effects. I would reword to "examine how the heating profiles of the simulated storm change".*

We have simply changed this to "To examine how the simulated storm impacts the temperature profile, …"

*Pg. 11, line 7: "covering the whole parameter space" – is this the whole input parameter space?*

Yes, it is.

*Pg. 12, line 4: Instead of number density, should this be number concentration?*

As suggested, this has been changed to "number concentration" (also three lines above this occurrence).

*Pg. 12, line13: when referring to the "lowest number concentrations of hail", to what diameter are you referring?*

This statement referred to the whole size distribution. To clarify this, we have inserted this information in parentheses: "The lowest number concentrations of hail (over the entire size distribution) are found …"

*Pg. 12, last two sentences: From this figure, it appears to me CCN has a larger effect than the strength of the ice multiplication.*

We have emphasized $a_H$ and the ice multiplication factor because both are microphysical parameters and their impact is consistent over the entire size range except for the two largest diameters. The impact of CCN and $a_G$ is also large but varies significantly over the considered size range. This has now been expanded upon:

„The corresponding plot of the main effect (Fig.3, right) confirms the impact of the fall velocity of hail ($a_H$) and the strength of the ice multiplication together to be responsible for large parts of the output uncertainty of the number concentration at all considered diameters except at D>25 mm. These two parameters contribute more than 50 % to the output uncertainty for these diameters. At the largest considered diameters, an increased contribution from the CCN concentration comes into play, while smaller diameters are also impacted by the graupel fall speed. This may be linked to the two formation pathways of hail in COSMO, namely through freezing of rain (of which the size is impacted by the CCN concentration) and through riming of graupel."

*Pg. 13, Table 3: Per Khain et al. 2011 (Atmospheric Research), the signal of CCN changes associated with hail fall switches sign around 3000 cm-3. Do the authors think their results are in line with this study?*

No, as shown and discussed in Wellmann (2018, Fig. 7.13 therein), our results are not in line with Khain et al.'s (obtained from a 2D simulation of a model with a bin microphysics scheme), but rather with Noppel et al (2010), who used also the COSMO model. However, we think that this discussion is out of the scope of this manuscript.

*Pg. 14, lines 3-4: I would argue the contribution due to CCN is larger in both S1 and S2 than S3.*

The reviewer is correct for the precipitation variables, but not for the hydrometeor contents. However, our statement that S3 is more similar to S2 than to S1 except for cloud water and snow is also correct. We have thus kept the sentence at it is, and have added tables with the numerical values in the appendix (Tables A1 and A2).

*Pg. 15, lines 34-40: See specific comment #2.*

As indicated above, we have emphasized the importance of the input parameter range once more in the abstract and conclusions. In this paragraph, we think that the two last sentences were already very clear and have been reformulated only slightly: "Furthermore, in our study the parameter range of the wind shear is chosen to reflect typical forecast errors and not a broad range of atmospheric conditions. This results in a smaller impact of the wind shear variation compared to the setup of Dennis and Kumjan (2017)."

*Pg. 16, line 10: See specific comment #2.*

Here we point out that our results are not directly comparable to the results of Storer et al. (2010) because of different input parameter ranges. We don't see any necessity to change this statement.

*Pg. 15, second line 5: it seems like the authors are arguing there is a difference between "the cloud" and "the integrated amount of cloud water" in the Storer et al. (2010) study. Could they explain what his difference is?*

This statement has been clarified: "In their study, the updraft strength and the total accumulated precipitation are very sensitive to changes in CAPE, while the integrated amount of cloud water does not depend strongly on CAPE."

*Pg. 16, Fig. 5: I'm unable to see the different standard deviation distributions. Perhaps instead a similar plotting method as in Fig. 6, with individual lines of different styles marking the edges of the standard deviations.*

We have added horizontal bars for the standard deviation at one selected altitude.

*Pg. 17, lines 28-29: Saturation adjustment was not one of the input parameters selected to test, so how can the authors make this claim?*

Our sentence was misleading. We have rephrased this to: "Condensation of cloud water, which is a substantial contributor to the total heating rate in the lower and middle troposphere, is parameterized via a saturation adjustment scheme in our model. Nevertheless, it yields a large contribution to output uncertainty of the diabatic heating in all three setups. This effect might be even larger if a time-dependent treatment of condensation was used."

*Pg. 17, lines 3-4 (just before section 5.3): See specific comment #2.*

We have removed this sentence.

*Pg. 18, lines 17-18: Can this sentence be worded more clearly? Perhaps "the spread of the distributions in S2 is larger than S1, particularly for smaller diameters of hail."*

To also satisfy reviewer #1, who also had a comment on this sentence, we have reformulated it to "For S2, the low (dashed red line) and high (continuous red line) hail size distributions are smaller and larger, respectively, than those for S1, leading to a larger spread in the distributions."

*Can the authors comment why the uncertainty is so much larger for smaller diameters?*

This statement is unclear to us. The spread is rather smaller for smaller diameters, not larger.

*Pg. 19, first and second lines: The authors note the relationship between low fall velocity of graupel and high number concentrations of hail several times in the paper. Could they provide a physical explanation for this relationship?*

We have added tentative explanations:

"low fall velocities of graupel (presumably resulting in more time for riming of graupel and growth to hail) and high fall velocities of hail (possibly by leaving less time for melting below the cloud) lead to high number concentrations."

*Pg. 19, line 20: "than the inputs related to environmental conditions". . .on the scale of uncertainty seen in COSMO. See specific comment # 2.*

We have added "(with the spread of input parameters chosen in this study)".

*Pg. 21, line 23-26: An excellent summary of the potential impact of this research – but it depends on the transferability of the results to other microphysical parameterizations (see specific comment #1).*

We agree with the reviewer and have added the following paragraph (as mentioned above): "It can be expected that our results (in particular regarding the microphysical parameters) depend to some extent on the microphysics scheme of our model. However, the overarching aim of this study was not to emphasize the impact of a specific parameter, but to quantify the relevance of environmental versus microphysical uncertainty in general. We expect that these results are less dependent on the microphysics scheme."

*Typographical: Pg. 4, line 31: Add a comma after profile. Pg. 10, Eq. 3: instead of nu/s, which denotes division, I'd use nu,s.*

All of these have been corrected.

*Pg. 10 line 34: "such that"–> where*

This would change the meaning of the sentence. However, we decided to omit the second half of this sentence entirely, because it doesn't give any new information.

*Anonymous Referee #3*

*Review of "Comparing the impact of environmental conditions and microphysics on the forecast uncertainty of deep convective clouds and hail" by Wellmann at al.*

*The study identifies model input parameters describing environmental conditions and cloud microphysics that lead to large uncertainties in the prediction of deep convective clouds and precipitation, by conducting statistical emulation and variance-based sensitivity analysis of the simulated deep convective clouds in an idealized setup of a cloudresolving model. They showed some interesting results that could be useful in guiding the improvement of forecasting. However, the results could be very dependent of microphysics scheme, model setup (such as idealized vs. real, nesting vs non-nesting), and even convective case. This discussion would be necessary. Particularly, the two moment microphysics schemes with saturation adjustment for condensation/evaporation calculation could lead to very different CCN impacts on latent heating and precipitation rate compared with more explicit microphysics schemes such as bin scheme as detailed in a review article (Fan et al., 2016, JAS). This could change the conclusion related to the diabatic heating rate. Another major problem of this manuscript is that the authors only described what the figures show, and did not interpret the results by connecting with physics properties/processes. See my specific comments for Section 4 and 5. The paper also have quite a bit confusing statements that need to be clarified. Therefore, a major revision is recommended to improve the paper before it is accepted for publication.*

We agree that the details of the parameterizations employed here have the potential for a significant impact on the results. However, our main aim was not to emphasize the impact of a specific parameter, but rather to weigh the relevance of environmental versus microphysical uncertainty. We expect that this result is less dependent on the microphysics scheme.

We have added the following paragraph to the conclusions: "It can be expected that our results (in particular regarding the microphysical parameters) depend to some extent on the microphysics scheme of our model. However, the overarching aim of this study was not to emphasize the impact of a specific parameter, but to quantify the relevance of environmental versus microphysical uncertainty in general. We expect that these results are less dependent on the microphysics scheme."

Furthermore, we agree with the reviewer that the saturation adjustment scheme in our model may lead to an overestimation of condensation and latent heating. The uncertainty related to condensation may be underestimated. This is now stated more explicitly in the conclusions: "Condensation of cloud water, which is a substantial contributor to the total heating rate in the lower and middle troposphere, is parameterized via a saturation adjustment scheme in our model. Nevertheless, it yields a large contribution to output uncertainty of the diabatic heating in all three setups. This effect might be even larger if a time-dependent treatment of condensation was used."

Also regarding the expected dependence on the case and setup, we agree with the reviewer. However, more complex setups would have been computationally more expensive, and also more difficult to analyse. To our knowledge, no study exists in which emulators have been developed for real or nested cases. Therefore we think that an extension of this work into the direction of more and more complex cases would be desirable, but is out of the scope of the present manuscript. We have added "This rather simple setup was required to allow a large number of simulations in which environmental conditions and microphysical parameters are modified."

And in the last paragraph: "In addition, future studies should address how far the results of our idealized simulations are transferable to real cases."

*Specific comments*

*The title has a grammatic error: it should be "Comparing . . . to (or with). . ."*

According to Merriam-Webster, "comparing apples and/to/with oranges" all are acceptable. Reformulating the title as suggested would make it significantly longer, and we have therefore decided to keep it as it is.

*Abstract:*

*Need some detailed background about how change of environmental conditions affect deep convective cloud properties. P1, L8-9 I think the results section showed that fall speed of graupel even contributes more than the fall speed of hail. In the last sentence, suggest rewrite or add sentence to show what parameters impact hail.*

We have rewritten the second but last sentence and now explicitly mention the role of the graupel fall speed. Regarding more detailed background information, we think that this is better placed in the introduction than in the abstract.

*Introduction:*
*1. P1, L16, Fan et al., JGR, 2009 and Qian et al., JGR, 2015 are the studies focusing on wind shear impacts on convective clouds.*

We have added the suggested references, but note that this list was not meant to be exhaustive.

*2. P2, L5, Change the second "for" to "with".*

Done.

*3. P2, L22-23 The sentence "However, the impact on precipitation is not identified as the investigated clouds are non-precipitating" needs to be rewritten. I have no idea what you want to say here.*

We meant that the clouds simulated by Igel and van den Heever (2017) were non-precipitating and therefore, this study made no statements on the impact of the cloud drop size distribution shape parameter on precipitation. We have removed this sentence and inserted two words in the previous sentence: "Igel and van den Heever (2017) vary the shape parameter of the cloud droplet size distribution in simulations of non-precipitating shallow cumulus clouds."

*4. P2, L25, The sentence "because of its higher fall velocity immediately falls out of the cloud leading to reduced convection intensity" has grammar errors and also confusing.*

This sentence has been reformulated to "Their results show that ``hail-like'' (large and dense, with a high fall velocity) graupel immediately falls out of the cloud, leading to a reduced convection intensity."

*5. Need to change the strong tone in some statements., for example,*
*(1) "There are only a few studies including Lee et al. (2008) and Storer et al. (2010) where the effect of several parameters is analyzed", you do not need to say only a few studies since there are a significant number as far as I Know. If indeed just a few, all of them are needed to be cited here. (2) "The only previous studies of multiple interacting uncertainties in deep convective clouds are our own previous study (Wellmann et al., 2018) and Johnson et al. (2015)."*

In the first of the mentioned sentences, we have changed "only a few" to "a few". Regarding the second statement, we believe that this is correct, but we have clarified that we refer here to studies with multiple (six or more) interacting parameters. With a few other reformulations, this paragraph now reads:

"The development of deep convective clouds is sensitive to both environmental conditions and model parameters, but these sensitivities are usually examined separately. A few studies, including Lee et al. (2008) and Storer et al. (2010), have analyzed the effect of several parameters, yet the maximum number of considered parameters is three or less. In this study, we combine various parameters related to both environmental conditions and microphysics into a single comprehensive sensitivity analysis. In idealized high-resolution model simulations, the selected input parameters are modified and their effect on the model output is analyzed with a special focus on precipitation and thermodynamic quantities. To our knowledge, the only previous studies of multiple (six or more) interacting uncertainties in deep convective clouds are our own previous studies (Wellmann et al., 2018; Johnson et al., 2015)."

*Model Setup:*
*1. Since the open lateral boundaries are used, need to specify how the boundaries are set up, i.e., what are used for the boundary conditions?*

We specify this a couple of sentences later:

"The initial temperature and humidity profiles (which are also used when air is advected into the domain through the boundaries) are based on those of Weisman and Klemp (1982)."

*2. P4, L20 How did you define cloudy points?*

This has been added:
"where the vertically integrated content of any hydrometeor type is >0"

*3. P4, L34-35, the recent progress about CCN impacts on convective clouds is Fan et al., (2018, Science).*

Instead of adding a 10th reference for this rather simple statement, we have inserted "e. g." and reduced the number of references to 3. Our original list was by no means meant to be exhaustive.

*4. P5, L15-16, this is confusing, how can you specify the wind velocity to be constant in all simulations since wind is a prognostic field?*

This refers only to the initial profile. This has been clarified.

*Sections 4 and 5:*
*1. I think some brief introduction to the case is needed before discussing the results from uncertainty quantification (UQ), which would help understand the UQ results . For example, I would like to know the relative amount of each hydrometeor mass to understand if this is a hailstorm case or not (i.e., hail mass is dominant compared with graupel mass). This would help me understand why graupel fall speed is the largest contributor to the uncertainty of integrated hydrometeor mass.*

As our ensemble encompasses a wide range of parameter values, it is not possible to describe "the" case. One example realization is depicted in this figure from Barrett et al. (2019). Most cases have more graupel than hail aloft, but this reverses at lower levels. We think that the reason why graupel fall speed is so important lies in the formation pathway of hail by riming of graupel.

[Figure]

*2. In both Section 4 and 5, there is a problem that the authors only describe the figures, but do not interpret the results from physics perspective. For example, in describing Fig. 1, it is better to understand why graupel fall speeds and CCN have the largest impacts on integrated hydrometeor mass but not on the hail mass? Why CCN have a large contribution to integrated hydrometeor mass but not to diabatic heating?*

We agree with the reviewer that more discussion of the physical processes was necessary. However, it is difficult to derive causal relationships from our statistical analysis. Therefore, most explanations are speculative, and we have indicated this wherever necessary.

Regarding the CCN impact on hydrometeor contents, we have added: "The second most important parameter is the CCN concentration, which contributes especially to the uncertainties of cloud water (because it determines autoconversion and thus impacts the partitioning between cloud and rain water) and snow content."

And in section 4.2: "CCN contributes only modestly to uncertainty at these levels, although the heating rate by condensation is very strong here. This is probably linked to the fact that a saturation adjustment scheme is used for cloud condensation, which is thus insensitive to droplet number and size."

*3. I have a hard time to physically understand the contributions shown in Figure 2. At the maximum heating around 3-6 km, the latent heating should be dominated by condensation, which should be strong affected by CCN. Bout because saturation adjustment is used for condensation and evaporation, the CCN effect on condensation is not shown here. This problem should be discussed.*

See above.

*In addition, How does graupel and rain fall speed contribute to the heating uncertainty? Above 10 km, the major contributors are CCN and graupel fall speed. I think it is because they affect how much amount of condensate mass are being transported to the upper levels. This kind of discussion is important to connect with cloud physics.*

We agree with the interpretation of the reviewer. The text we have added reads: "Above [10 km], the output uncertainty of the total heating rate is dominated by the CCN concentration and the fall velocity of graupel. This is probably linked to the indirect effect of CCN and riming efficiency on the amount of supercooled water transported to the homogeneous freezing level. Furthermore, graupel is produced at these levels in our model as a result of the freezing of rain drops, and the graupel fall speed factor thus impacts the gravitational sink of the (small) graupel particles present at these altitudes."

*4. P12, L18-20 Figure 3 shows the largest contribution is graupel and hail fall speeds, which is different from what is described here.*

In this sentence, we have emphasized the parameters which are important for the entire size range of hail. The graupel fall speed is important only for the smaller diameters. The paragraph has been reformulated to clarify this and to add some physical explanation:

"The corresponding plot of the main effect (Fig. 3,right) confirms the impact of the fall velocity of hail ($a_H$) and the strength of the ice multiplication together to be responsible for large parts of the output uncertainty of the number concentration at all considered diameters except at D< 25 mm. These two parameters contribute more than 50 % to the output uncertainty for these diameters. At the largest considered diameters, an increased contribution from the CCN concentration comes into play, while smaller diameters are significantly impacted by the graupel fall speed. This may be linked to the two formation pathways of hail in COSMO, namely through freezing of rain (of which the size is impacted by the CCN concentration) and through riming of graupel."

*5. P12, L20-21, need to discuss the possible physical mechanisms of how CCN affect the large hailstones. There are literature studies about this.*

We are unsure which references the reviewer refers to. As obvious in our answer to item 4, we refer here to the hail formation processes in our microphysical scheme.

*6. P15, the first three paragraphs, need some discussion in connecting with cloud physics to understand why. For the third paragraph, how to explain the contrasting contribution of hail fall speeds to hydrometeor mass and precipitation?*

The first paragraph now includes a statement on the possible mechanism for the suppression of sensitivity to INP: "Thus, the main effect of the INP concentration is smaller if other microphysical parameters are used as input, possibly because other ice phase processes (secondary ice formation, riming) can suppress the sensitivity of a cloud to primary ice formation."

The third paragraph has been revised as follows: "When looking at the hydrometeor masses, the contribution from the fall velocity of hail to the output uncertainty is negligible except for the integrated hail and rain contents. However, it is the largest contributor to the uncertainty of the precipitation variables, presumably reflecting that hail itself and melted hail constitutes a major part of the total precipitation."

*7. P17, "Dennis and Kumjian (2017) specify in their work that process rates are not an essential factor causing discrepancies in the formation of hail for different model setups", not sure what this means, since microphysical process rates directly determine the budget.*

We agree that this sentence was confusing, and have removed it.

*8. Section 5.3, need clearer introduction here to state the purpose of this part. I was not understanding the purpose of this part until I got to the summary (last paragraph of page 20).*

We have moved part of the last sentence of p. 20 to the beginning of section 5.3: "In this section, we analyze the impact of variations of environmental conditions and microphysical parameters on the size distribution of surface hail."

*9. P18, L12-21, all of the magnitudes described in these two paragraphs are different from what is shown Figure 6. For example, the maximum value plotted is 0.01 mm-1m-3, but you got values of 0.4 and 3.4 mm-1m-3 in the text. Need to check what is going on.*

We apologize for this mistake. The numbers in the text were wrong and have been corrected.

*10. P18, L15-16, the sentence is confusing and need clarifications.*

For clarification, this sentence has been split into two sentences: "For this setup (in which the environmental conditions are modified), the controlling parameters are the CCN and INP concentrations and $\theta_0$ . Low number concentrations of hail arise for higher values of these parameters and high number concentrations of hail for lower values."

*Section 6:*
*1. The relevant summary (the first three paragraphs) needs to be revised accordingly by adding physical explanations.*

As the summary mostly describes microphysical and environmental parameters as a package, we therefore think that iterations on our thoughts on the underlying processes for the effects of individual parameters would not be well placed here.

Where CCN and graupel fall speed are mentioned, we have added "These parameters are crucial for the efficiency of warm and cold rain formation, respectively."

*2. For "The controlling parameters of the combined input parameters are the INP concentration and the fall velocities of graupel and hail, hence a combination of parameters describing environmental conditions and microphysical parameters", the logic of the sentence is wrong. All the parameters described here are only microphysical parameters.*

We removed "hence a combination of parameters describing environmental conditions and microphysical parameters" to avoid confusion, but note that in our setup S1, INP concentration is counted as an environmental parameter because it characterizes the aerosol environment (different from the other microphysical parameters, which refer to uncertain microphysical parameterizations).

[revised manuscript text omitted]

---

## Author Response (AR2)

We would like to thank the editor and reviewer #3 for the careful reading and useful comments. We have addressed all of them and inserted our answers below. The reviewers' comments are in italics, while our answers are in normal font. Where we refer to new/modified text from the manuscript, this is given in blue color.

In addition, we have
- changed present tense to past tense in the description of previous studies
- added one sentence to the abstract to stress the most important quantitative results:
"In particular, the uncertainty in the fall velocities of graupel and hail account for more than 65% of the variance of all considered precipitation variables and for 30-90% of the variance of the integrated hydrometeor mass contents."

***Editor comments***

*Editor comment 1) Your response to the following comment by Referee #2 was great. However, adding this information to the text would help clarifying Figure 6. Please add the information of the first and second sentence of your response (maybe in a rephrased form) to the manuscript.*

*Referee #2: 4. The discussion of the variability of the hail size distribution caused by the different input parameters focused solely on the maximum and minimum number concentrations and not the distribution of responses within those bounds. Within Fig. 6, could box and whisker plots be used to show the distribution of number concentrations within each of the three setups within a set range of size bins? That would allow the distribution of distributions, as it were, to be discussed.*
*Author response: For Fig. 6, the spread is based on 64 parameter combinations with characteristic high and low values, not on 10,000 evenly distributed combinations. The reason for this difference to Fig. 5 is that our aim was to attribute the minimum and maximum hail size distributions to specific parameter distributions. Therefore box and whisker plots would not be well suited here.*

Thanks for this suggestion. We have added these sentences in section 4.2, because this is where the corresponding analysis is discussed for S2:

"The size distribution of surface hail is simulated using the emulators for all possible combinations of the high and low input parameter values for each setup (128 combinations in S2, 64 combinations in S1 and S3). The aim of this approach is to attribute the minimum and maximum hail size distributions to specific parameter combinations."

*Editor comment 2) I noticed that you changed 'hydrometeor mass' to 'hydrometeor mass contents' throughout the manuscript. However, you missed a few instances. Please replace all 'hydrometeor mass' and 'hydrometeor content' to 'hydrometeor mass content'.*

Thanks for pointing this out. We have replaced this everywhere now, we hope.

*Editor comment 3): p. 1, l.2: 'CCN contributes only moderately...'. Do you mean 'CCN concentration...'? – Please specify.*

Corrected.

*Editor comment 4): p. 1, l.4: 'Cloud condensation' is a rather uncommon term and only used in the context of 'cloud condensation nuclei'. Better would be 'water vapor condensation'.*

Corrected as suggested.

*Editor comment 5): p. 1, l.21; As you changed in the line above and three lines later 'number concentration' to 'number density', this should be here used too*

The previous changes were actually in the other direction (changing "number density" to "number concentration", so we have corrected this for the third instance, too.

*Editor comment 6): p. 20, l. 6: Do you mean 'hailstone concentration' or 'hailstone mass' by 'amount of hailstones'?*

We meant the number concentration, more precisely the size-resolved number concentration. This sentence has been rewritten to clarify this:

"When both the environmental conditions and the microphysics are perturbed, the lower limit of the size-resolved number concentration of hailstones approximately doubles compared to S1."

*Editor comment 7): Tables A1 and A2 are not referred to in the manuscript. If there content is relevant, please mention them in the manuscript text.*
*If they are not part of the main text, they can be either in an Appendix or in the Supplement. Please look at the 'manuscript preparation guidelines' for the formatting.*
*https://www.atmospheric-chemistry-and-physics.net/for_authors/manuscript_preparation.html*

The tables were added in response to a reviewer comment, because the graphical depiction in Fig. 4 is hard to read quantitatively. We have now added a reference to the tables in the caption of Fig. 4. The tables are intended as an appendix; however, the appendix title ("Appendix A: Numerical values" ) unfortunately appears at a funny place at the end of page 24, although we followed the instructions in the latex template. We hope that this problem can be solved during typesetting.

*Referee #3*

*The authors addressed most of my comments. Here are a few further minor comments based on their responses:*

*1. About my 1st comment in Sections 4 and 5: The authors did not indicate address the comments in the manuscript. The type of figure and information of "Most cases have more graupel than hail aloft, but this reverses at lower levels. We think that the reason why graupel fall speed is so important lies in the formation pathway of hail by riming of graupel" should be included in the manuscript.*

Thanks for this comment. We have decided against including a figure with the vertical profiles of hydrometeor content for two reasons: First, such a plot can only depict one ensemble member, and would not be representative for the whole ensemble. Second, similar figures have been published in other papers, and it seems more economic to save space here by referring to one of those. We have therefore added a reference to Wellmann et al. (2018) which included a description and analysis of the simulations in S1, and brief description in Section 2 (bottom of page 4):

"Exemplary vertical and horizontal cross sections of the idealized convective cloud simulated with this configuration are shown in Wellmann et al. (2018, their Fig. 3). Typically, the cloud contains more graupel than hail at upper levels, but hail persists longer below the melting level and (in addition to rain) only hail, not graupel, reaches the ground."

As for the hail and precipitation formation pathways, they are discussed several times later in the manuscript, e.g. on page 13:

"As shown by Barrett et al. (2019), roughly half of the surface rain in this model setup originates from cold rain processes involving riming. Therefore here the graupel (and also hail) fall speed parameters contribute substantially to the uncertainty of the latent heating rate at levels below 2 km, although there is no graupel present at these altitudes."

*2. The sentence added to address my 2nd comment in Sections 4 and 5 "The second most important parameter is the CCN concentration, which contributes especially to the uncertainties of cloud water…" is not accurate. One of the most important effects of CCN on cloud water is through enhancing condensation, as emphasized in many studies such as Sheffield et al., JGR, 2015, Fan et al., Science, 2018.*

In the context of our model (and the sentence cited by the reviewer is taken from results section), this statement appears to be correct. In the microphysics scheme used here, condensation is not affected by CCN (because of the saturation adjustment scheme), but the effect of CCN is expressed through autoconversion. To stress this, we have added "in the microphysics scheme used here":

"The second most important parameter is the CCN concentration, which contributes especially to the uncertainties of cloud water (in the microphysics scheme used here, primarily via an impact on autoconversion and thus on the partitioning between cloud and rain water) and snow content."

*3. About CCN impacts on hail size, here are the example studies I meant: Loftus and Cotton, AR, 2014 and Khain et al., AR, 2011.*

Thanks for the suggestions. We have added the following sentence:

[revised manuscript text omitted]